# A machine learning approach for online automated optimization of super-resolution optical microscopy

Audrey Durand[1], Theresa Wiesner[2], Marc-André Gardner[1], Louis-Émile Robitaille[1], Anthony Bilodeau[2], Christian Gagné[1], Paul De Koninck[2,3] & Flavie Lavoie-Cardinal[2]

Traditional approaches for finding well-performing parameterizations of complex imaging systems, such as super-resolution microscopes rely on an extensive exploration phase over the illumination and acquisition settings, prior to the imaging task. This strategy suffers from several issues: it requires a large amount of parameter configurations to be evaluated, it leads to discrepancies between well-performing parameters in the exploration phase and imaging task, and it results in a waste of time and resources given that optimization and final imaging tasks are conducted separately. Here we show that a fully automated, machine learning-based system can conduct imaging parameter optimization toward a trade-off between several objectives, simultaneously to the imaging task. Its potential is highlighted on various imaging tasks, such as live-cell and multicolor imaging and multimodal optimization. This online optimization routine can be integrated to various imaging systems to increase accessibility, optimize performance and improve overall imaging quality.

[1] Département de génie électrique et de génie informatique, Université Laval, Québec, QC G1V 0A6, Canada. [2] CERVO Brain Research Center, 2601 de la Canardière, Québec, QC G1J 2G3, Canada. [3] Département de biochimie, microbiologie et bio-informatique, Université Laval, Québec, QC G1V 0A6, Canada. Correspondence and requests for materials should be addressed to A.D. (email: audrey.durand@mcgill.ca) or to F.L.-C. (email: flavie.lavoie-cardinal.1@ulaval.ca)

Super-resolution techniques have revolutionized the field of optical microscopy by overcoming the diffraction limit and allowing the observation of protein dynamics at the nanoscale[1,2]. Among these techniques, point scanning methods such as STimulated Emission Depletion (STED)[3] and REversible Saturable Optical Linear Fluorescence Transition (RESOLFT)[4] are particularly well suited for the study of dynamical processes in living cells or multicolor imaging[5–7]. However, as they are further developed, along with the continuous evolution of bio-imaging tools and experimental paradigms, these super-resolution methods come with several layers of complexity in their implementation in the laboratory[8–10]. Such complexity calls for the development of automation strategies to increase accessibility of super-resolution microscopy for live-cell, multicolor, and multi-modal imaging.

The success of the imaging process depends on the tuning of many parameters, such as laser excitation and depletion power, pixel size, scanning speed, detector gating, and illumination scheme. Concretely, this success is characterized by the evaluation of different, possibly conflicting, objectives (e.g. improving resolution and signal to noise ratio (SNR), while reducing photobleaching, light exposure, and imaging time). Furthermore, the range of relevant imaging parameters may be greatly influenced by the biological structure, the fluorophores, the experimental conditions, and the biological variability. Gaining experience with these experimental settings requires considerable time and resources, discouraging many from using super-resolution microscopy for complex tasks, such as live-cell imaging.

We propose here an online machine learning approach to improve the performance of optical nanoscopy by addressing an online optimization problem, where the aim is to maximize the outcome (here the objectives) during the real imaging phase. We formulate this problem under the multi-armed bandits framework[11] and tackle it using the Kernel TS[12] approach, which combines posterior sampling with kernel regression to efficiently model each objective. We achieve multi-objective (MO) optimization by first including an expert into the optimization routine that is providing feedback on the quality of acquired images and on the trade-offs that can be made among the different objectives. Next, we propose to remove the user from the loop using artificial neural networks to automate the overall image quality evaluation[13] and to articulate the expert preference among optimization objectives. This combined approach results in a fully automated system for optimizing imaging parameters online. We show that complex techniques such as STED, multimodal microscopy, or live-cell optical nanoscopy strongly benefit from a fully automated parameter optimization. This strategy could be implemented on a wide range of microscopes, without restriction to super-resolution, to conduct the optimization simultaneously with the imaging task. This strategy should increase the efficiency of the imaging process and standardize the results.

## Results

**Single-objective optimization of imaging parameters**. Traditional, offline optimization techniques search for good parameters in a prior exploration phase (also known as pure exploration), followed by the phase of imaging (also known as exploitation). Online optimization algorithm, on the other hand, acquires images with different parameters, as it explores the parameter space, and is learning which regions in the parameter space result in better or worse images while simultaneously attempting to produce suitable images. This results in the so-called exploration-exploitation trade-off[14], which is the main scope of the well-known multi-armed bandits framework[11]. Here, we consider a bandits algorithm combining kernel regression and Thompson

Sampling (TS)[15], namely Kernel TS[12], to capture the underlying structure of the parameter space and efficiently model each objective.

In order to illustrate the potential gains of conducting an online optimization compared with a conventional pure exploration approach, we first considered optimizing one imaging parameter toward one objective (Supplementary Figure 1). More specifically, we designed a replay experiment (see Methods) to compare the proposed Kernel TS-based approach with a basic grid search (GS, a commonly used approach in biology) on the same set of images, removing the effect of variability throughout biological samples. To this end, we generated a dataset of 468 STED images of the axonal periodical actin lattice, a biological structure only detectable with optical nanoscopy, as the periodicity of actin rings is approximately 180 nm[16] (Fig. 1a). The images were acquired at 12 evenly distributed laser excitation power values. Briefly, the replay experiment consisted in simulating the imaging at a given parameter by sampling, from the dataset, images that had been acquired previously using this parameter. To quantify the success of the imaging process, we evaluated different measurements that were compatible with online analysis on a standard computer: the autocorrelation amplitude of the actin periodical pattern[17], the SNR, the photobleaching, and the Fourier Ring Correlation (FRC)[18,19] (Fig. 1b and Supplementary Figures 2 & 3). With the excitation power as the parameter to optimize, we considered the autocorrelation as the objective to maximize (Fig. 1c). More specifically, we defined the relevant parameter region as four powers between 4.0 and 9.7 μW, corresponding to maximal values of the resulting autocorrelation target function (Fig. 1c).

Each replay trial consisted in 60 images, i.e. 5 images per laser power for GS. In the case of Kernel TS, the number of images per laser power varied depending on the exploitation-exploration trade-off and the resulting confidence on the different regions of the parameter space. This resulted in more selections of excitation power values associated with a higher autocorrelation amplitude $(4.0-9.7\,\mu W)$ (Fig. 1d and Supplementary Figure 4). We characterized the performance of our optimization algorithm by counting the number of imaging failures, the success being the ability to distinguish clearly the axonal periodical lattice on an image (Fig. 1e). This is related to the regret measure used to characterize bandits optimization algorithms, which must be minimized (in opposition to success) (Supplementary Figure 5). While an oblivious algorithm would lead to a linear (tendency) regret along time, learning algorithms are known to achieve sublinear (e.g. square root or logarithmic) regret[20]. Results showed a clear sublinear trend in the average regret curve for Kernel TS in comparison to the linear trend obtained with GS. In each of the 100 replay trials, we compared the absolute deviation between the estimated and targeted autocorrelation for the most often selected parameter by Kernel TS (Fig. 1f). As expected, by allocating more samples to potentially better parameters, especially for a measure varying strongly with the chosen biological region (Fig. 1c), Kernel TS was able to provide a better estimate of the autocorrelation objective function compared with GS (Fig. 1f). These results support the benefits of online parameter optimization and the applicability of Kernel TS for the task and, more importantly, highlight the benefits of online optimization.

Diverse studies have shown that factors, such as resolution, detected signal, imaging speed, and even the type of imaged structures strongly influence the resulting imaging quality[21–24]. Quality assessment of super-resolution images has been performed in an offline fashion using dedicated software[21]. Here, we rely on a scoring system of the overall image quality rated by an expert user based on the signal, the noise level, and the observed

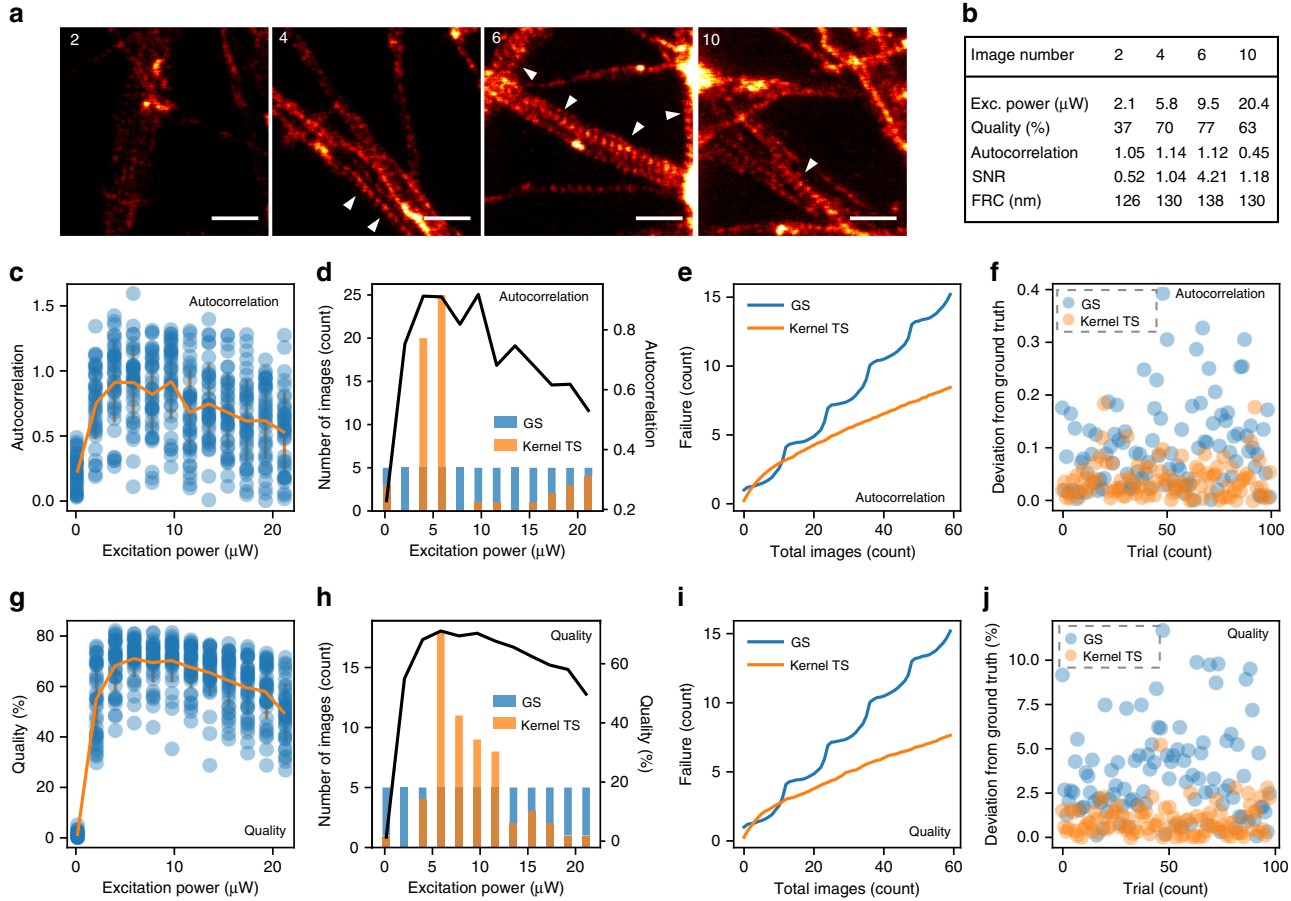

**Fig. 1** Replay experiment of single-objective optimization. **a** Selected images of actin, labeled with phalloidin-STAR635 in fixed cultured hippocampal neurons, from the first ten images of the dataset at four increasing laser power values. Scale bars 1 μm. White arrowheads: actin periodical lattice. **b** Comparison between the quality scores, SNR, autocorrelation of the periodical actin lattice, and FRC measurements obtained for the images shown in **a** and acquired at the indicated excitation power. **c**, **g** Autocorrelation (**c**) and image quality (**g**) values measured for all images of the dataset (blue dots) and average of the 39 autocorrelation (**c**) or quality (**g**) values per excitation power (orange line). **d**, **h** Distribution of the selected excitation power for 60 images obtained in one replay trial for both GS and Kernel TS when evaluating the autocorrelation (**d**) or the image quality (**h**). The black line indicates the averaged functions (orange line in **c** and **g**) for autocorrelation and for image quality. **e**, **i** Average cumulative regret of imaging failures (not detecting the actin periodical lattice) when optimizing the autocorrelation (**e**) or image quality (**i**). Note that the wave pattern is due to the decreasing probability of obtaining good images as parameters move further away from the relevant region. **f**, **j** Error between the estimated and ground truth autocorrelation (**f**) or image quality (**j**) values at the excitation power that was most often selected by Kernel TS for each optimization trial. **g**, **h** Quality scores from 0 to 1 are expressed in percentage

resolution, allowing several objectives to be jointly considered. With our neuronal samples, this quality rating was found to be less dependent on biological variability than autocorrelation and could better characterize images with very low signal levels. Indeed, in comparison to flat, rounded cells, neurons provide less uniform signals, due to the complex and highly variable shapes of neuronal processes, which may explain why classical approaches, such as SNR evaluation or FRC, sometimes fail to evaluate the image quality (Supplementary Figures 2 & 3). We updated our dataset by manually rating the 468 super-resolution images (Fig. 1g). The relevant parameter region for the quality remains the same as for the autocorrelation. As described above for the autocorrelation objective, we performed replay experiments, using image quality as feedback to Kernel TS. The cumulative regret curve (Fig. 1i) still displayed a sublinear trend. By allocating more samples to relevant parameters (Fig. 1h), an online optimization approach (hereby Kernel TS) is able to provide a better estimate of the objective function at the most often sampled parameter (Fig. 1j), compared to an offline strategy (hereby GS). These results indicate that a quality score provides a

fast and reliable measurement for online optimization of super-resolution images.

**Multi-objective optimization of super-resolution images**. To obtain optimal super-resolution images, multiple objectives must be considered, many of which having potential conflicts. Thus, several imaging parameters have to be optimized by making trade-offs. This calls for an efficient MO and multi-parameter optimization strategy. We tackle this task by (i) relying on independent instances of Kernel TS to model each objective function and (ii) letting an expert provide feedback about its preference articulation among objectives, thereby guiding the MO optimization process (Methods and Supplementary Figure 6). In the proposed MO optimization setting, the benefits of online optimization are highlighted by comparing the Kernel TS-based approach with offline optimization using the Non-dominated Sorting Genetic Algorithm (NSGA-II) (that is better suited than GS for MO optimization)[25]. NSGA-II searches the parameters space in order to identify the parameter configurations lying

anywhere on the Pareto front, that is the ensemble of all non-dominated trade-offs between objectives (i.e. solutions for which there are no other better solutions over all objectives)[26]. Simulation and experimental results comparing the performance of NSGA-II and Kernel TS optimization are shown in Supplementary Figures 7, 8 and 9. As expected from an offline approach, NSGA-II is able to identify multiple trade-offs, but requires a large number of poor images to achieve this result. Though other online optimization approaches (e.g. GP-UCB[27], Kernel UCB[28]) could be considered in the single-objective optimization setting, Kernel TS is especially well-suited for the described MO optimization setting as it can generate options that are informative about the underlying objective function (see Methods), rather than maintaining upper confidence bounds or other similar indices.

One frequently faced challenge in super-resolution microscopy, when addressing a new biological question, is to adapt the imaging parameters to a new fluorophore. We investigated if Kernel TS could be used in such a case by comparing the optimization routine when using three far-red emitting dyes (ATTO647, STAR RED, and Alexa633) for STED imaging of the protein $\alpha$-tubulin in fixed neurons. ATTO 647 and STAR RED are known to be good fluorophores for STED nanoscopy due to their high quantum yield and photostability[29]. To our knowledge, due to its low photostability[30], no work featuring Alexa633 with this technique has been published, even if it is a very common fluorophore for confocal microscopy. An optimization sequence consisted of searching within 768 possible parameter configurations (pixel dwelltime, excitation and depletion laser power) using only 100 images, while pursuing two conflicting objectives: maximizing the image quality and minimizing photobleaching. The parameter hyper-volume was kept constant, and we evaluated the most frequently selected parameter combination for each fluorophore (Fig. 2b). For all fluorophores, the optimization process allowed to achieve high image quality (Fig. 2a). Even if the results obtained for photobleaching and image quality were comparable for ATTO647 and STAR RED (Fig. 2a, c), the parameter combination leading to these results were different (regions featuring blue and orange circles in Fig. 2b). We also observed that high-quality images could be obtained with Alexa633, but at a lower rate than with the other two dyes (Fig. 2c). In contrast to the other dyes, very low excitation powers (15-fold lower than for ATTO647) were required for Alexa633 in order to control the photobleaching and therefore achieve the targeted trade-offs (Fig. 2b). These results show that our MO optimization routine can effectively provide optimal parameters to select with different fluorophores in order to obtain better images. Additional results for characterizing the reproducibility and comparison with manual optimization by a human expert are provided in Supplementary Figures 10 and 11.

Next, we assessed the performance of the proposed MO optimization approach in the context of high biological variability and limited number of measurements (images) per sample. We performed live imaging of an intracellular protein involved in synaptic signaling, $\alpha$ Ca$^{2+}$/calmodulin-dependent protein kinase II ($\alpha$CaMKII)[31,32], in dendrites and spines of cultured neurons. With super-resolution microscopy, $\alpha$CaMKII should appear as a dense array of discrete puncta[33]. One major challenge in super-resolution imaging of living cells is to reduce the photobleaching and phototoxicity associated with repeated imaging and high intensity illumination, while preserving high resolution and signal[34]. The optimization was conducted by simultaneously varying the laser excitation power, the laser depletion power, and the pixel dwelltime. An optimization trial consisted of searching among 343 possible parameter configurations, using only 80

images due to the limited imaging time per live sample (Fig. 2d). Three objectives were considered: (i) the quality of the first STED image, (ii) the quality of the second STED image, and (iii) the photobleaching. We observed that the available trade-offs between the objectives and the optimal parameter space were highly variable between the samples, either day to day or across coverslips, even though each contained the same protein of interest ($\alpha$CaMKII-SNAP-SiR) (Fig. 2f and Supplementary Figures 12 and 13). We observed an improvement of the image quality along the optimization process (Fig. 2e and Supplementary Figure 14), while the photobleaching could be kept below 50% after two STED images. These results indicate that despite the high biological variability of the preparation under study, online optimization could help in improving the robustness of the imaging process.

We also assessed the suitability of our method to other commonly used cell types and probes by evaluating the performance of online Kernel TS-based optimization on: (1) various GFP-tagged structures in live neurons, Human Embryonic Kidney (HEK293) and pheochromocytoma (PC12) cells and (2) variable staining intensities of the actin cytoskeleton in fixed neurons. Using fluorescent proteins for STED imaging poses an additional challenge for time-lapse live imaging due to photobleaching. Although the label was the same (GFP) for many targeted structures, we observed that the (i) day-to-day biological variability, (ii) cell type, (iii) level of expression, and (iv) density of the target protein strongly influenced the imaging parameters and the outcomes (image quality, photobleaching, imaging speed) (Fig. 3a and Supplementary Figures 15–31). The most frequently sampled parameters to obtain good trade-offs between the objectives varied greatly across cell types expressing LifeAct (Fig. 3a, Supplementary Figures 15–31). For the three presented optimization sequences, good trade-offs between objectives were achieved (Supplementary Figures 21, 22, 24–27, 29, 30), which resulted in a sublinear accumulation of low-quality and high photobleaching images (Fig. 3b). Detailed results showing multiple optimization sequences for each presented structure and cell types as well as for other GFP-tagged proteins are presented in Supplementary Figures 15–31. In all cases, online optimization improved the quality of STED images acquired from living cells.

One might expect that previous knowledge from an optimization sequence acquired on a different sample could accelerate the optimization process. We tested for this possibility with live-cell imaging and quantified the proportion of good quality images with low photobleaching and imaging time. We analyzed the results obtained with previous knowledge generated from optimization sequences (100 images) (i) of a standard reference sample for super-resolution microscopy (GATTAQUANT Nanobeads), and (ii) of a transfected postsynaptic marker (PSD95-FingR-GFP) (Supplementary Figures 15–17)[35]. We observed that previous knowledge could not reliably improve and accelerate the optimization process depending on cell types, structures, and day-to-day variability. For example, using the previous knowledge from GATTAQUANT Nanobeads OG488 negatively impacted the performance when imaging proteins in neurons and HEK-cells (Supplementary Figures 18, 19, 21–24, 26, 27), while positively impacting when imaging LifeAct-GFP in PC12 cells (Supplementary Figures 29 and 30). We also tested if previous knowledge would accelerate the optimization process by reducing the hyper-volume of optimization parameters. For this experiment, we applied MO optimization to a common problem in cell imaging, the variability in labeling density. We performed Kernel TS optimization (100 images) on two fixed neuronal samples in which the actin cytoskeleton was stained with (1) low (1:1000) and (2) high (1:100) concentration of phalloidin-STAR635

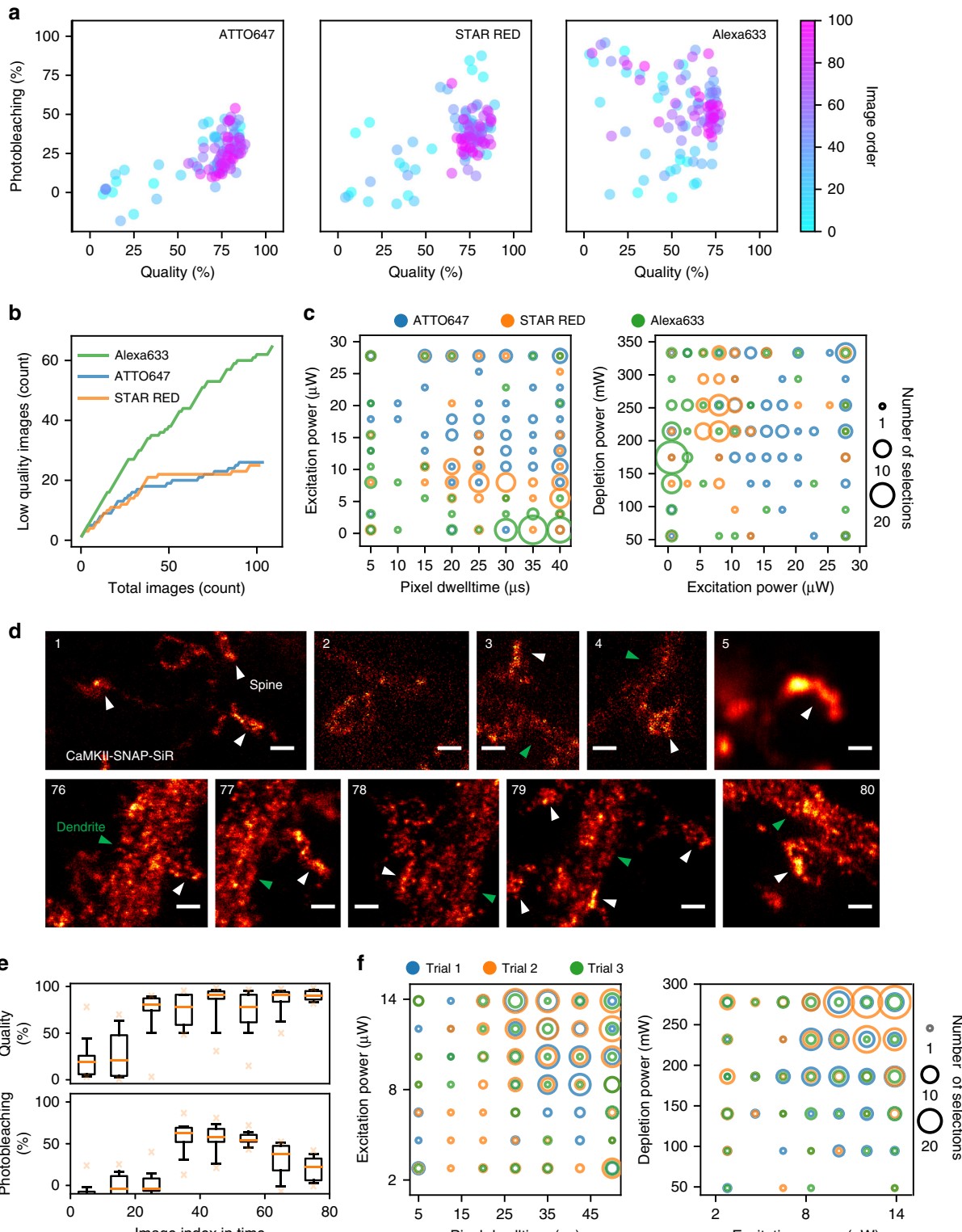

**Fig. 2** Multi-objective optimization of STED imaging on fixed and live neurons. **a** Evolution of the objectives during an optimization sequence, from first (cyan) to 100th image (pink), for the protein α-tubulin marked with ATTO647, STAR RED, and Alexa633. **b** Cumulative regret (related to the quantity of low-quality images, i.e. below 70%) obtained during the Kernel TS optimization sequences shown in **a**. **c** Parameter configurations selected by Kernel TS for fluorophores shown in **a**. **d** Example images of αCaMKII tagged with SNAP-SiR of the first five (1–5) and last five (76–80) images taken during one live-cell optimization trial. White and green arrowheads highlight the position of dendritic spines and shafts respectively. Scale bars 500 nm. **e** Distribution of image quality and photobleaching for one optimization sequence of αCaMKII-SNAP-SiR (Trial 2 in panel **f**). One bin corresponding to the average of ten images. Orange line indicates the median, box covers the first to the third quartiles, and whiskers extend from 10th to 90th percentiles. **f** Parameter configurations selected by Kernel TS during different live-cell imaging trials. In **c** and **f** the size of the circles scales with the number of images that were acquired with a given configuration. Shown are two planes of the three-dimensional parameter space. Quality scores from 0 to 1 are expressed in percentage

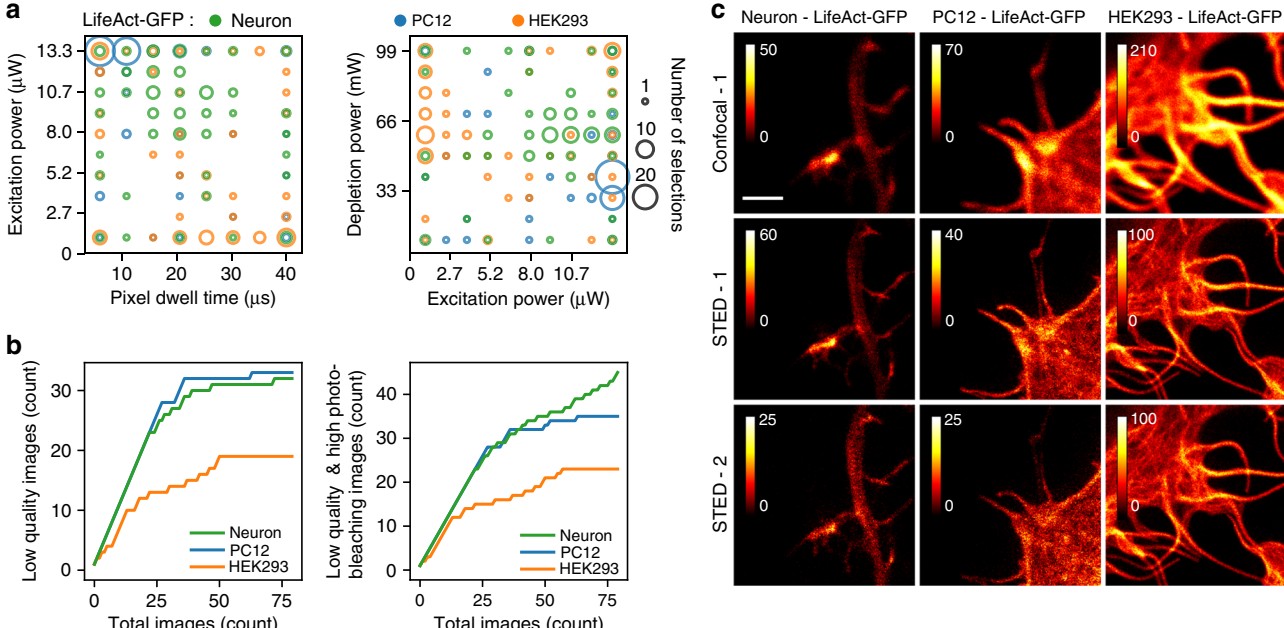

**Fig. 3** Multi-objective live-cell optimization of GFP imaging. **a** Parameter configurations selected by Kernel TS during different imaging trials in three different cell types: neurons (green), PC12 (blue), and HEK293 (orange). **b** Cumulative regret curve of (left) image quality alone (images with a quality score below 60%) and (right) image quality and photobleaching (images with a quality score below 60% or photobleaching above 75%). **c** Example images obtained among the last ten images of one optimization sequence for each cell type. The confocal image was taken before two consecutive STED images (labeled as STED-1 and STED-2). Note the differences in intensity scales across images to reflect differences in fluorescence intensity (confocal images) or photobleaching (STED1 vs. STED2). Scale bar 1 μm

(Supplementary Figures 32, 33). We reduced the hyper-volume of imaging parameters using the regions that were sampled more often during the full optimization sequence to perform two new optimization experiments on the low dye concentration sample. When using the reduced hyper-volume from the high dye concentration, high-quality images with low photobleaching were obtained for only 38% of the images (19 out of 50 images with Quality > 60% and Photobleaching < 60%) and the actin rings were detected in only 22 of the 50 images (Supplementary Figure 32e). In contrast when using the reduced hyper-volume resulting from optimization on the same sample, the trade-off between objectives resulted in better trade-offs (39 out of 50 images with Quality > 60% and Photobleaching < 60%) and actin rings were clearly visible on 48 out of 50 images (Supplementary Figure 32e). These results emphasize that prior knowledge to reduce the hyper-volume of parameters is not always reliable for optimizing super-resolution microscopy and depends on the difference between the chosen biological samples. By adapting to the staining quality, online optimization can help to detect structures that could otherwise have been missed with non-optimal imaging parameters.

As a last illustration of the benefits of the proposed online MO optimization approach based on Kernel TS, we complexified the task by combining optimization of local glutamate uncaging in living neurons with parameter optimization of STED microscopy (Fig. 4a–c). Multimodal approaches combining optical stimulation and fluorescence microscopy have been widely used to correlate morphological changes and neuronal activity. The goals of this experiment were to acquire STED images of the NMDA receptor subunit GluN2B[36] (Fig. 3c) and to elicit a local synaptic response, restricted inside an identified dendritic spine, using focal MNI-Glutamate photo-uncaging near the spine head. To monitor the synaptic response, we measured $Ca^{2+}$ fluctuations revealed by the genetically encoded $Ca^{2+}$ indicator GCaMP6s[37] (Fig. 4a, b). We aimed to optimize the laser power and distance from the spine of the

uncaging spot, with the objectives of minimizing the local spread of the response (response size ratio) and maximizing the intensity of GCaMP6s response ($\Delta F/F$ peak intensity). The response size ratio was calculated as a ratio of the region where a $Ca^{2+}$ transient was detected and the whole dendritic area (see Methods). A response was considered to be local if the response size ratio was non-zero and below a threshold, referred later as the response size threshold. A random sampling (RS) of parameters was conducted as a baseline of comparison with Kernel TS (Fig. 4e). Due to high biological variability between samples regarding the parameters and the responsiveness of the neurons (Supplementary Figures 34 and 35), images obtained during several trials were considered together in a bootstrapping analysis (see Methods). While local responses were observed using RS, their relative frequency compared to widespread responses was significantly lower than with Kernel TS (Fig. 4f, Supplementary Figure 36, and Methods). According to the posterior regression model, only a very tight range of parameter configurations led to local responses (Fig. 4d). Kernel TS increased the success rate of obtaining a localized $Ca^{2+}$ response (Fig. 4f) and found well-performing parameters to be very variable between the samples (Supplementary Figure 34). As a trend, we observed that local stimulation could be performed with moderate power (20–30 μW) at 500 nm from the spine head, which corresponds to distances regularly used in such experiments[38] or alternatively with distances up to 2 μm by increasing the uncaging laser power (35–40 μW) (Fig. 4d, e). Likewise, a high variability in the nanoscale reorganization of GluN2B in spines following the photostimulation was observed and will be the subject of future investigation.

As expected, when considering multimodal imaging, the success probability of achieving all objectives without optimization is very low (Fig. 4f, g). Indeed, a threefold increase was observed in the frequency of success (simultaneous local uncaging response and high STED image quality) between the beginning (mostly random exploration) and the end of the imaging process driven by optimization (10.7−29.4%) (Fig. 4g). These results

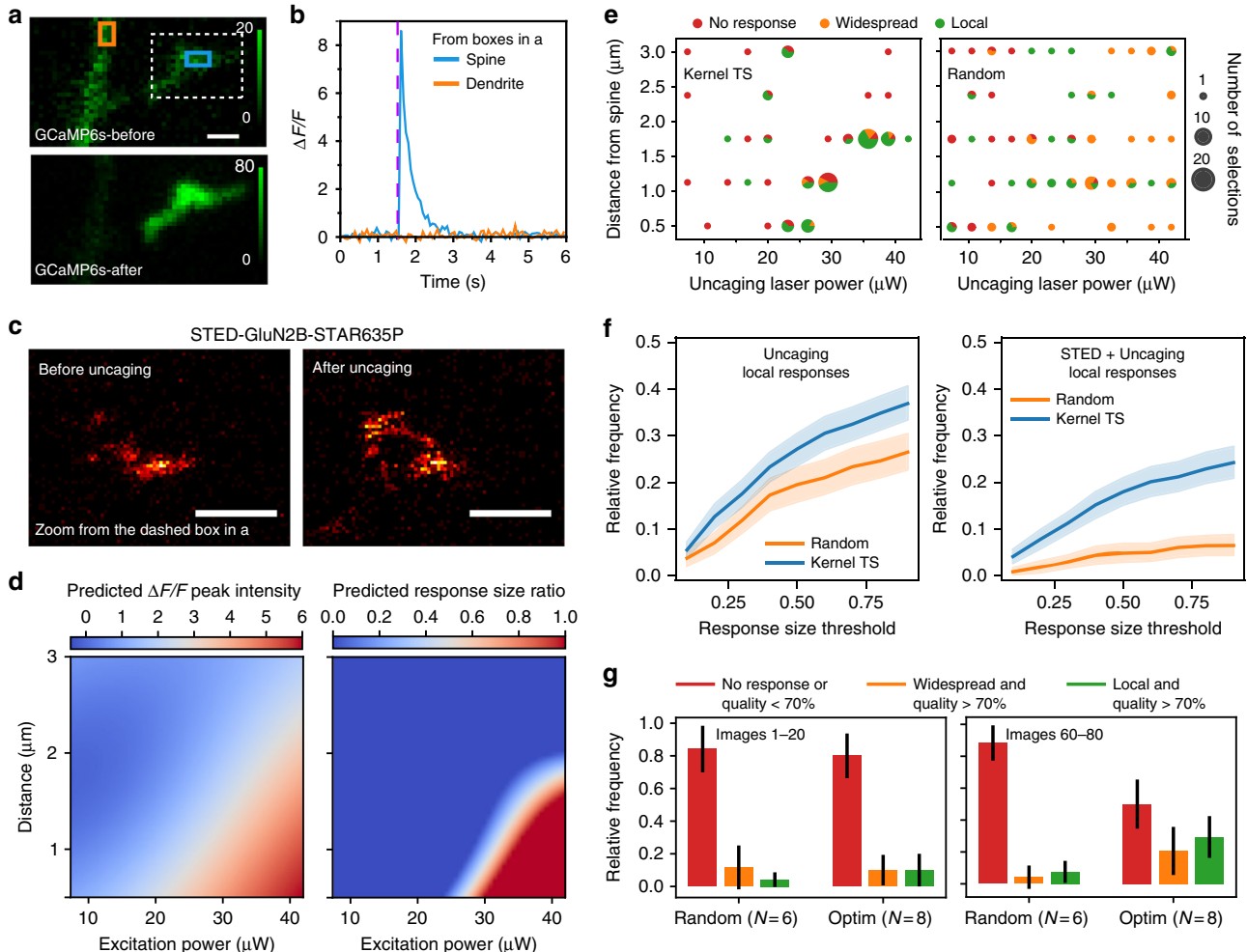

**Fig. 4** Multimodal optimization in living cells. **a** GCaMP6s fluorescence before and after glutamate uncaging. **b** Change in Ca²⁺ concentration inside the ROIs in spine (cyan rectangle in **a**) and dendrite (orange rectangle in **a**). Ca²⁺ response was limited to the spine. **c** GFP-GluN2B tagged with FluoTag-X4 anti-GFP STAR635P before and after uncaging (zoomed inset from the dotted box in **a**). Scale bars 500 nm. **d** Objective functions, i.e. ΔF/F peak intensity (left) and response size ratio (right), predicted by kernel regression after 80 observations. The colorbar indicates the predicted value for this objective, in the objective units. **e** Types of Ca²⁺ responses after glutamate uncaging for the selected parameters during one optimization (left) and random sampling (right) trials. Pie charts show the repartition between response types, i.e. local (green), widespread (orange) and no response above threshold (red). The size of each pie chart is proportional to the number of trials of the parameter configuration. **f** Relative frequency of local response ratios over all images obtained in several trials of 80-image sequences, for different response size ratio threshold values. 95% confidence intervals are obtained by 1000 bootstrap repetitions. **g** Comparison of the success rate of multimodal experiments using RS and Kernel TS for the images 1–20 and 60–80. Vertical bars indicate one standard deviation computed over the N repetitions. With Kernel TS, a significant increase in the frequency of local responses and decrease in the no-response rate are observed between the first and the last images (Kruskal−Wallis, p = 0.01 and p = 0.004 respectively), with RS no significant difference in the frequency of local response and no-response rate was observed between the first and the last images (Kruskal−Wallis, p = 0.45 and p = 0.57 respectively). **e**, **g** The threshold for the response size ratio was set to 0.6

show the benefits of using online parameter optimization when tackling multifaceted task that would require considerable and advanced experience, or extremely long sequences of attempts to identify the right parameters to use.

**Fully automated optimization of STED imaging parameters.** While being effective for online optimization, as shown above, quality rating is characterized by a high variability between users and can be influenced negatively by decision fatigue[39]. Thus, automating the quality rating should further improve and speed up the optimization process by removing the need for manual quality rating. It could in fact improve both the accessibility to super-resolution microscopy for less experienced users and the standardization of results over time.

Deep neural networks were successfully applied to automate image quality assessment[13], by being able to learn which relevant features to extract rather than defining them through a careful algorithms handcrafting, like it is required with classical methods. While promising, this approach[13] was limited to images of fixed dimensions containing structures of a fixed scale. Here we propose a fully convolutional neural network (FCN) architecture able to cope with images of arbitrary dimensions and resolutions, facilitating transfer to a new task. This FCN (Fig. 5a) receives a STED image as input and outputs a score map, assigning a quality score to each pixel given its surroundings at various scales (Fig. 5b)[40]. Score maps can then be averaged to a global score between 0 and 100%, mimicking an expert rating. The score maps provided by the FCN (Fig. 5b, Supplementary Figures 37 and 38) are informative as they help motivating the inferred score. With them, the network

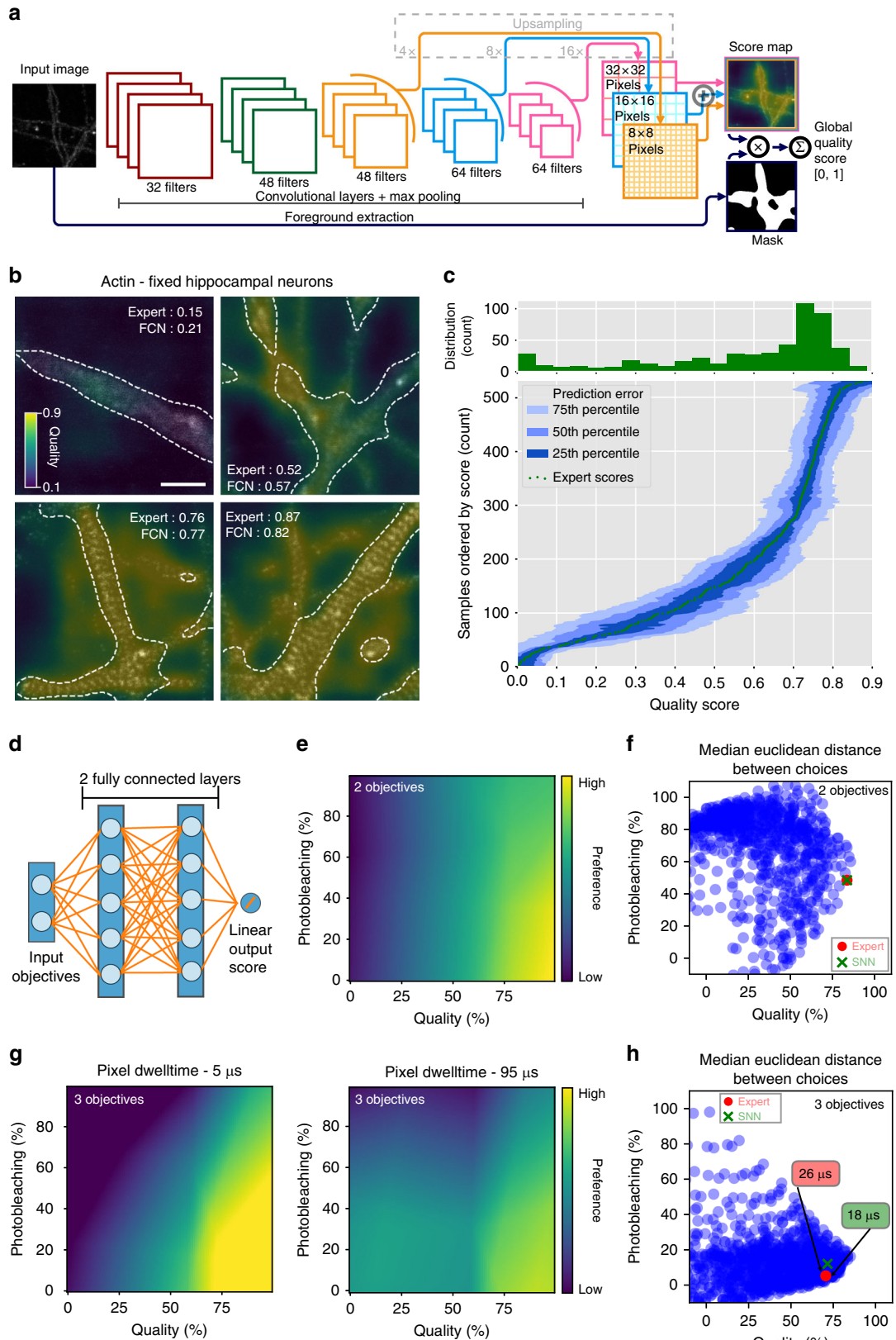

can precisely point out which region of the image contributed to its resulting score. This constitutes a crucial step in the process of understanding the behavior of such FCN models, and may assist in training new users in evaluating microscopy images.

We trained the proposed FCN model on various datasets of either fixed or living hippocampal neurons (see Methods and

Supplementary Figure 39). The performance of the proposed FCN model was assessed by comparing its ratings with quality scores provided by an expert user on a test dataset of similar nature (Fig. 5c, Supplementary Figure 40). Investigating the possibility of fine-tuning a pre-trained network using a small amount of data from a different protein or structure (see

**Fig. 5** Automatic image quality rating with an FCN and automatic preference articulation with an SNN. **a** Proposed FCN architecture for quality rating. Each convolutional layer is followed by spatial batch normalization and an exponential linear unit (ELU) activation. All layers but the last one are also followed by maxpooling with kernel 2 × 2. The output is a linear combination of the three last layers, allowing to handle different sizes of features in the image (see Methods). **b** Example of score maps produced by the FCN for images of the actin cytoskeleton stained with phalloidin-STAR635. Scale bar 1 μm. **c** Distribution of qualities in the validation sets (top histograms) and error distributions of an FCN trained on the Actin dataset, on test images of the actin cytoskeleton stained with phalloidin-STAR635. The green dots are the expert scores, sorted by value. The blue areas represent the extent of the errors made by the network at every level of quality. **d** Architecture of the SNN for preference articulation, given two objectives as input. Each fully connected layer contains ten neurons. **e** Preference articulation function between photobleaching and image quality learned by the two-objective SNN model. **f** Example of choices made by the two-objective SNN vs. an expert based on the median error given by Euclidean distance (see Methods). **g** Preference articulation function between photobleaching, image quality and total imaging time learned by the three-objective SNN model. **h** Example of choices made by the three-objective SNN vs. an expert based on the median error given by Euclidean distance (see Methods). **e–h** Quality scores from 0 to 1 are expressed in percentage

Methods), we observed that it is possible to achieve good performance in predicting quality scores, using less than 100 images for training (Supplementary Figure 41), which makes the approach practical even with costly image acquisition processes. We also demonstrate a different application, where the FCN is asked to rate quality of images acquired with a widefield microscope (see Methods and Supplementary Figure 42). This supports the possibility to extend the proposed model to another microscopy environment. We released code and training data for this deep network and also provided access to trained models so that our experiments can be replicated and extended (see Methods).

In the MO optimization setting, the expert is presented multiple options (one per parameter configuration), referred to as a cloud of options, sampled by Kernel TS in the objective space. The next parameters are selected based on the preference articulation among these options (see Methods). Since the preference function describing its trade-offs strategy on the whole objective space cannot completely be explained, the choice of options cannot be addressed using a trivial rule-based or parametric approach. For this reason, we relied on a shallow fully connected neural network (SNN) (Fig. 5d) to learn the preference articulation function from expert choices. In opposition to the FCN, which is made of convolutional layers applying filters on their input, the SNN is made of fully connected layers applying dot products between their input and learned weights (Fig. 5a, d). Here, the SNN is trained to regress a score function (Fig. 5e, g) in the objective space by using pairwise comparisons (see Methods and Supplementary Figure 43). Given a cloud of options provided by Kernel TS, the idea is to predict the score of each option using an SNN and select the option with the highest score. We trained two SNN models: one for articulating the preference between two objectives (image quality and photobleaching) and one for articulating the preference between three objectives (image quality, photobleaching, and time per pixel). Both SNN models were trained on option clouds and choices gathered during previous, user-driven, optimization runs of fixed and live-cell imaging. We compared the choices made by an SNN with expert choices on the same clouds, and observed that the SNN choices are very close to the expert choice in all objective dimensions (Fig. 5f, h).

We combined the three modules (Kernel TS, FCN, and SNN) into a fully automated (FA) MO optimization platform for imaging parameters. Options sampled from Kernel TS (one model per objective) were provided to the SNN to identify the best trade-offs and select its corresponding imaging parameters to be used by the super-resolution microscope. The obtained image was scored with an FCN and the resulting quality score, along with other objectives, were used as feedback to update the kernel regression models of the objective function (Fig. 6a). We evaluated the resulting fully automated system on different fixed and live-cell imaging tasks with various proteins and dyes.

We first optimized three parameters (pixel dwelltime, excitation, and depletion laser power) while maximizing imaging quality and minimizing photobleaching for the imaging of various structures in fixed hippocampal neurons (actin, tubulin, PSD95, Bassoon, and GluN2B). All experiments used the FCN trained on the Actin dataset (see Methods). Images acquired along optimization runs show that the fully automated system is able to improve without the help of an expert user (Fig. 6e and Supplementary Figure 44). Similarly to user-driven optimization, this translates into sublinear regret (Fig. 6b) and convergence to a trade-off in the objectives space (Supplementary Figure 45a). Note that the amplitude of the curves cannot be used to compare approaches together as they are influenced by stochasticity of the experimental conditions and biological variability. We also observe that this is achieved using different parameter regions depending on the structure/protein, supporting the need for dedicated optimization (Fig. 6c and Supplementary Figure 45b). These results show that the FCN trained on one protein (hereby Actin) is able to generalize well to other proteins. We evaluated if an FCN fine-tuned on less than 100 images could be successfully used in FA optimization. We considered FCN models (i) trained on the Actin dataset, (ii) fine-tuned on the Tubulin dataset, and (iii) fine-tuned on 96 images of the Tubulin dataset. All resulted in similar imaging parameter regions associated to the best objective trade-offs (Supplementary Figure 46). We then conducted three-parameter (pixel dwelltime, excitation, and depletion laser power) optimization while maximizing imaging quality, minimizing photobleaching, and minimizing imaging time during live-cell imaging (GFP-αCaMKII, PSD95-FingR-GFP, and LifeAct-GFP) (Fig. 6d). Each of these experiments used an FCN model fine-tuned for the protein of interest (see Methods). This allowed each FCN to specialize to the appropriate image resolution for each protein, a benefit of the FCN architecture. Images acquired along optimization runs show that the fully automated system is able to improve without the help of an expert user (Fig. 6e and Supplementary Figure 44). This translates into convergence to different objectives trade-offs that are relevant given the task at hand: the system is able to (1) improve the quality while controlling the photobleaching and pixel dwelltime for LifeAct-GFP; (2) maintain the quality level while reducing the photobleaching and pixel dwelltime for PSD95-FingR-GFP; and (3) improve the quality while controlling the photobleaching, at the price of increasing the pixel dwelltime for GFP-αCaMKII (Fig. 6d and Supplementary Figure 47). This is also achieved using different parameter regions depending on the structure/protein, supporting the need for dedicated optimization (Supplementary Figure 48).

These results show that the image quality rating provided by an FCN constitutes a valid input to Kernel TS, and that preference articulation made by an SNN is able to drive the optimization process for finding well-performing parameters. This framework could be implemented on various microscopes, for example in a microscope facility, to allow non-expert users to maximize their

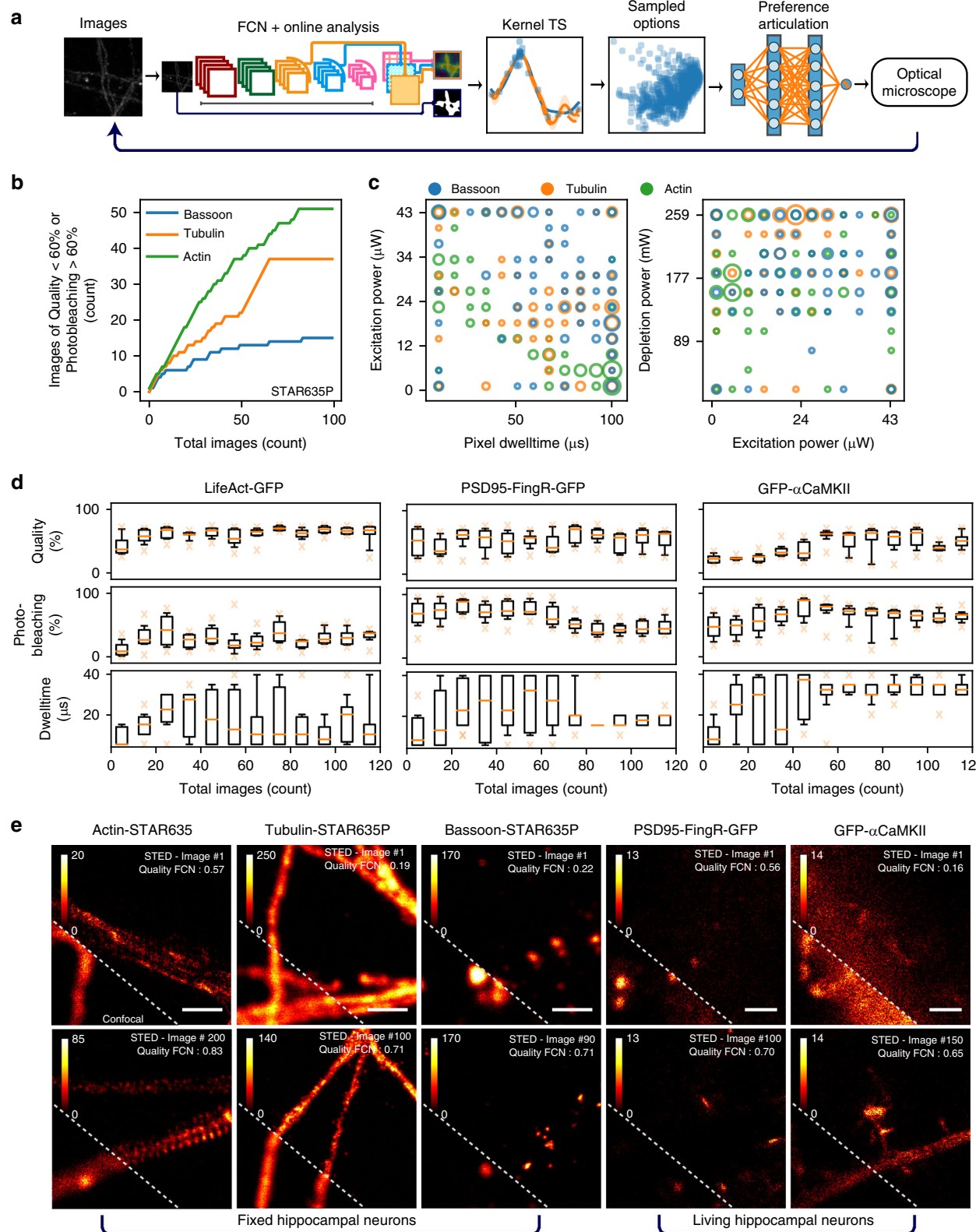

imaging results, without previous knowledge on well-performing parameters for their particular experiment.

## Discussion

We have proposed an online optimization approach, MO Kernel TS, for optimizing several imaging parameters jointly while conducting an imaging task. We combined it with neural network approaches for image quality recognition and user preference evaluation to develop a fully automated optimization platform for super-resolution microscopy. Multi-objective Kernel TS optimization contrasts with traditional approaches where optimization is based on a pure exploration phase prior to the imaging task (known as offline optimization). Our results from various

**Fig. 6** Fully automated optimization system for STED imaging. **a** Scheme of the fully automated system where both networks interact with Kernel TS to automatically select the next imaging parameters (SNN) and rate the quality of the obtained images (FCN). **b** Cumulative regret (related to the quantity of low-quality images, i.e. below 60%, or high photobleaching, i.e. over 60%) obtained during FA optimization on fixed neurons stained with phalloidin-STAR635 (actin), bassoon-STAR635P and tubulin-STAR635P. Note: The regret is computed using quality scores given independently by an expert on the resulting images. **c** Parameter configurations selected by Kernel TS during optimization using Bassoon, Tubulin, and Actin proteins. **d** Evolution of the objective values during FA optimization sequences of live-cell imaging. Each box corresponds to the binned objective values of ten images. For LifeAct-GFP FA optimization improved the quality while controlling the photobleaching and pixel dwelltime. For PSD95-FingR-GFP it allowed to maintain the quality level while reducing the photobleaching and pixel dwelltime. In the case of GFP-$\alpha$CaMKII it improved the quality while controlling the photobleaching, at the price of increasing the pixel dwelltime. Quality scores from 0 to 1 are expressed in percentage. **e** Images acquired without human intervention during FA optimization sequences on fixed and live neurons. The dotted line delimits the confocal (lower left) from the STED (upper right) images. Scale bar 1 μm

experimental contexts and analysis methods indicate that the proposed machine learning approach for online optimization of super-resolution microscopy contributes to improve and standardize results across a wide range of samples and imaging modalities.

We first showed that by replacing a prior parameter exploration phase with an online optimization conducted simultaneously as the main imaging task, we were able to acquire significantly more high-quality images using well-performing parameters. We then showed that, though several objectives could be represented as a unified imaging quality score, MO optimization remained necessary when pursuing conflicting objectives. We also showed that well-performing parameters could vary from one fluorophore to another, as expected, but also from one sample to another. One could argue that the online optimization might lead to a misconception of the variance in image quality due to experimental treatment. However, the argument can be reversed by considering that a low-quality (or low intensity) staining combined with non-suitable acquisition parameters could lead to a wrong interpretation of the biological information. Indeed, if a user would optimize the imaging parameters on a given sample and always reuse them without adapting, wrongly chosen parameters could give the impression that a feature is not visible just because it is not detected. As an example, we showed how this issue applies to the detection of actin rings on samples of different staining qualities.

One common limitation of kernel regression is that it does not scale well with the increasing number of images in a sequence and the dimensionality of the parameters space. This is usually not an issue in the online setting, where the number of samples is limited, and so must be the search space. However, if one wishes to conduct optimization sequences over larger number of images (i.e. more than 10,000) to handle larger parameter spaces (i.e. more than four dimensions), it is possible to rely on recent works[41] to circumvent this issue. One should also note that the number of images required for finding a (near-)optimal solution scales with the number of effective dimensions in the search space, since the sharing of information decreases as the number of dimensions increases[27,28]. This is not a limitation of the proposed approach, but an inherent limitation of minimal space coverage.

We additionally showed that an FCN can be trained to rate quality of images according to specific tasks. This is done to capture the rating process of a human expert for some given tasks into a computer model, directly from a database of examples. This is an important element to be used for guiding Kernel TS in order to achieve successful online optimization of super-resolution microscopy. Combining this with an SNN to articulate preferences in the objective space allows the implementation of a fully automated optimization framework. This framework enables easier and faster optimization of imaging parameters that could now be performed on a daily basis and by non-expert users. This system could eventually be transferred to various imaging systems to increase accessibility, optimize performance and

improve overall result quality. The main limitation of the proposed neural network models resides in their need to retrain for completely new structures of interest (in the FCN case) or for different objective spaces (in the SNN case). However, our results suggest the seminal network trained on the Actin dataset to be very versatile and easy to adapt to other structures, capturing most of the structure intricacies and acting as an expert replacement even with very few training data. Also, given an important change of context that would require retraining either one or both networks, optimization could still be performed with a user in the loop, where useful data can be acquired while building a dataset in the same time. The provided open source code and detailed documentation should allow investigators to retrain the networks if necessary with their own datasets (see Methods).

Our proposal contrasts with current deep learning approaches for microscopy, which involve post processing of previously acquired images. Mainly, these approaches increase the image quality by improving resolution[42], denoising[43] or improving super-resolution image reconstruction[44], without addressing the efficiency of imaging acquisition nor reducing number of samples or data storage. Other works also combined confocal or two-photon microscopy with machine learning algorithms in order to achieve software-based aberration corrections with adaptive optics[45–47]. Imaging of subcellular structures in various types of living cells is complex, with significant biological variability between preparations, experimental conditions, samples, or even single cells. Kernel TS-based online optimization approach offers a useful solution for live-cell imaging, standardizing the quality of images between experiments. The proposed approach could also rely on different quality assessment scores[21] for guiding the optimization, given that they could provide feedback in an interactive fashion. In addition to the nonrestrictive use of our approach to different types of microscopy, it could be applied to the study of specific remodeling with optogenetics or nanoparticle-assisted localized optical stimulations, where a precise spatio-temporal control of the imaging modalities is necessary to tune synaptic functions[48,49].

The approach presented in this paper is a stepping stone toward intelligent nanoscopy. In the future, other constraints raised by the application should also be considered. For example, the capability of sharing knowledge gained through different trials and experiences could significantly improve the efficiency of the optimization process with a limited number of samples (e.g. on live samples) or large parameter space (e.g. multicolor imaging). Moreover, defining additional objectives in order to characterize other aspects of imaging, such as the ability to track changes in imaged structures, could push forward the contribution of optimization to live-cell imaging. This work demonstrates that machine learning offers many great possibilities to harness the power of leading-edge imaging devices.

## Methods

**Cell culture and transfections**. Dissociated rat hippocampal neurons were prepared as described previously[50,51]. Before dissection of hippocampi,

neonatal rats were sacrificed by decapitation, in accordance to the procedures approved by the animal care committee of Université Laval. Dissociated cells were plated on poly-D-lysine-coated glass coverslips (18 mm) at a density of 600 cells per mm$^2$. Neurobasal and B27 (50:1) were used for the growth medium supplemented with penicillin/streptomycin (50 U per mL; 50 µg per mL) and 0.5 mM L-GlutaMAX (Invitrogen). At the time of plating, fetal bovine serum (2%; Hyclone) was added. To limit proliferation of non-neuronal cells, half of the medium was changed without serum and with Ara-C (5 µM; Sigma-Aldrich), 5 days later. Thereon the culture was fed twice a week by replacing half of the growth medium with serum- and Ara-C-free medium. Pheochromocytoma PC12 cells and Human Embryonic Kidney HEK-293 cells were from the American Type Culture Collection. PC12 cells were maintained in ATCC-formulated RPMI-1640 Medium (#30-2001) supplemented with 5% fetal bovine serum (FBS-HI, Life Technologies, Inc.) and 10% horse serum (Horse Serum-HI, Life Technologies, Inc.). HEK-293T/17 cells were maintained in 10313-DMEM, high glucose, pyruvate, no glutamine (#10313021, Life Technologies, Inc.) supplemented with 10% fetal bovine serum and 4 mM Glutamax. One day prior to imaging, neurons were transfected using Lipofectamine 2000 (Thermo Fisher Scientific) with GCaMP6s[37] and αCaMKII-SNAP or GFP-GluN2B (gift from Stefano Vicini, Georgetown University) at DIV14-18. Same procedure was applied for neurons transfected with GFP-αCaMKII[50], PSD95-FingR-GFP (gift from Don B. Arnold, University of Southern California), membrane-GFP (farnesylated eGFP) obtained from Clontech (USA) or LifeAct tagged with monomeric GFP[52], while PC12 and HEK-293T/17 were transfected during the plating process.

**Fixation and immunostaining**. Cultured hippocampal neurons were fixed in freshly prepared 4% paraformaldehyde solution (4% sucrose, 100 mM phosphate buffer, 2 mM NaEGTA, pH 7.4) at room temperature for 20 min and washed three times for 5 min with phosphate buffer saline (PBS) (supplemented with 100 mM Glycine). Cells were permeabilized with 0.1% Triton X-100 and blocked with 2% goat serum (GS) for 30 min before immunostaining. Incubation with phalloidin, primary (PAB) and secondary antibody (SAB) was performed in a 0.1% Triton X-100 and 2% GS PBS solution at room temperature. PAB was incubated for 2 h at room temperature followed by three washes in PBS. SAB and phalloidin were incubated for 1 h and finally washed three times in PBS. The actin periodic lattice was stained with phalloidin-STAR635 (Abberior, cat. 2-0205-002-5, 1:50 dilution). The protein α-Tubulin was labeled with the PAB M-anti-α-Tubulin (Sigma-Aldrich, cat. T5168, 1:500) and one of the following SAB GAM-STAR635P (Abberior, cat. 2-0002-007-5, 1:100), GAM-ATTO647 (Sigma-Aldrich, cat. 50185, 1:250), GAM-STAR-RED (Abberior, cat. 2-0002-011-2, 1:250) or GAM-Alexa633 (Life Technologies, cat. A-21052, 1:250). The protein PSD95 was labeled with the PAB M-anti-PSD95(6G6-1C9) (Abcam, cat. MA1-045, 1:250) and the SAB GAM-STAR 635P (Abberior, cat. 2-0002-007-5, 1:250). The protein Bassoon was labeled with the PAB R-anti-Bassoon (Synaptic Systems, cat. 141 003, 1:500) and the SAB GAR-STAR635P (Abberior, cat. 2-0012-007-2, 1:250). Imaging of GluN2B in fixed hippocampal neurons was performed on neurons that were first transfected with GFP-GluN2B. After fixation, GluN2B was stained using FluoTag-X4 anti-GFP nanobodies (NanoTag, cat. N0304-Ab635P-L, 1:1000) for 1 h at room temperature. GATTAQUANT Nanobeads OG488 (size 23 nm) and GATTA-STED Nanorulers 70 nm labeled with ATTO 647N were obtained from GATTAquant GmbH Braunschweig, Germany.

**Cloning of αCaMKII-SNAP**. pCMV-SNAPf-αCaMKII was constructed by assembling the SNAPf tag from pENTR4-SNAPf (w878-1) (gift from Eric Campeau; Addgene plasmid # 29652) with the rat αCaMKII coding sequence. Briefly, the 546bf Nco1 (filled)-Xho1 SNAPf fragment was inserted to replace mGFP into the Nhe1 (filled)-Sal1 digested mGFP-αCaMKII[50].

**STED imaging, live-cell imaging, and glutamate uncaging**. Super-resolution imaging was performed on a four-color Abberior Expert-Line STED system (Abberior Instruments, Germany) equipped with four-pulsed (38–40 MHz) excitation lasers at 485, 518, 561, and 640 nm and two depletion lasers at 595 nm (38 MHz, 1 W) and 775 nm (40 MHz, 1.25 W). A ×100 1.4NA, oil objective was used and fluorescence was detected using four avalanche photodiode detectors (APD) with approximately 1 Airy unit detection pinhole. STED imaging of far-red dyes (ATTO647, Alexa633, SiR, STAR-RED, STAR635 and STAR635P) was performed with a 640 nm excitation laser, an ET685/70 (Chroma, USA) fluorescence filter, and a depletion donut situated at 775 nm (40 MHz). STED imaging of green dyes and fluorescent proteins (Oregon Green 488 (OG488), GCaMP6s, GFP) was performed with a 485 nm excitation laser, an ET525/50m (Chroma, USA) fluorescence filter, and a depletion donut situated at 595 nm (38 MHz). Scanning was conducted in a line scan mode with a pixel dwelltime of 5–100 µs and pixel size of 20–25 nm. In the far-red configuration we varied the laser excitation power from 2.8 to 13.7 µW (live cells) or from 0.65 to 26.8 µW (fixed cells). The laser depletion power was varied from 3 to 290 mW (live cells) or from 59 to 360 mW (fixed cells). For the green configuration, the laser excitation power was varied from 0.4 to 13.3 µW and the laser depletion power was varied from 9.7 to 99.3 mW. The laser power was measured in the back aperture of the objective. Our STED microscope was

equipped with a motorized stage and auto-focus unit. The imaging sequence was the following: (1) with a Python routine integrated in our optimization software, the user defined around 30 regions of interest of about 4 × 4 µm in a given imaging field; (2) those regions were then sequentially scanned without user intervention; (3) for each region a new combination of parameters was used; and (4) new options were shown to the user (or SNN) after each image to select the next imaging parameters. This imaging sequence was performed without the need for the user to move the stage or re-adjust the focus.

For super-resolution live-cell imaging of CaMKII-SNAP-SiR, neurons were pre-incubated with SNAP-SiR (3 µM, Biolabs) for 30 min and washed for a minimum 1 h in SNAP-SiR-free media prior to imaging. For super-resolution live-cell imaging of GFP-GluN2B, neurons were pre-incubated with the nanobody Fluo-Tag-X4-anti-GFP-STAR635P (2.5 nM, Nano-Tag Biotechnologies) for 1 h and washed for 30 min in fresh media. In order to minimize possible phototoxicity effects that could have long-term consequences on the imaging results, the considered range for both excitation and depletion powers was set not to exceed 30% of what is used with similar systems in the literature[53,54]. For imaging time, the range of pixel dwelltime values was set to obtain an image of 3 µm$^2$ (required to observe short dendritic segments and 1–3 spines) in less than 5 s. Since longer pixel dwell times were correlated with a blurring of the signal due to protein diffusion, the upper limit was set to 40 µs.

For super-resolution live-cell imaging of GFP-tagged structures, two STED images were acquired. The quality of the first image was evaluated by the user and the photobleaching was calculated using the foreground intensities of confocal images acquired before and after the STED stack. For the full Kernel TS optimization run, 80 images were acquired without the use of prior knowledge. It was followed by the acquisition of 2 sets of 20 images using prior knowledge from a full optimization run (100 images) on GATTAQUANT Nanobeads OG488 or on PSD95-FingR-GFP. For comparison with the images acquired with prior knowledge, the last 20 images of the optimization run without prior knowledge were used. In total, 120 images of the same sample were acquired. Only 20 images for each prior knowledge condition were acquired due to time limitations to preserve the healthiness of the cells under the microscope, which is inherent to live-cell imaging.

Imaging was performed in a closed circulation system with HEPES(4-(2-hydroxyethyl)-1-piperazineethanesulfonic acid)-buffered artificial cerebrospinal fluid (aSCF; Composition in mM: NaCl 104, KCl 5, HEPES 10, CaCl$_2$ 1.2, Glucose 10 and TTX 0.5; Osmolality: 240 mOsm per kg, pH: 7.3) at room temperature. For glutamate uncaging, MNI-glutamate (Tocris, Bristol, UK) was added to the bath at 0.5 mM and, using a peristaltic pump, the imaging solution was recirculated. Uncaging of MNI-glutamate was performed using a custom 405 nm laser path[55] (Supplementary Figure 49) allowing the independent control of the laser power and the uncaging distance parameters. The duration of uncaging illumination was 30 ms. Ca$^{2+}$ transients were acquired using time-lapse imaging at 4 Hz in confocal mode with 485 nm excitation and a 525/50 detection filter. For each optimization trial, 80 individual spines were chosen on 2−4 different neurons of the same coverslip. Clearly visible spines were identified on a confocal overview image of the chosen dendrite. We simultaneously varied the UV laser excitation power from 7.3 to 42.0 µW and the excitation spot distance from 0.5 to 3 µm from the spine head. HEK293 and PC12 cells were imaged in a closed circulation system with HEPES-buffered aSCF (Composition in mM: NaCl 120, KCl 5, HEPES 10, CaCl$_2$ 1.2, MgCl$_2$ 2, Glucose 10; Osmolality: 290 mOsm per kg, pH: 7.3) at room temperature. Control of all imaging parameters, imaging sequences, region selection, and online image analysis was performed using a custom-written python script and the Imspector-Python interface SpecPy (Abberior Instruments, Germany). Note that this automatic control of the microscope is possible because of the scripting option accessible with Aberrior microscopes. Image postprocessing was performed with the open source software Fiji-ImageJ2[56,57]. All presented data are raw data.

**Online analysis**. The glutamate uncaging optimization was performed with two different objectives: (i) minimizing the area of detected Ca$^{2+}$ response and (ii) maximizing the amplitude of Ca$^{2+}$ transients. This was performed by the following online detection routine. Two different stacks of images were acquired during the uncaging experiment: a baseline stack (25 frames) before the UV light pulse and a response stack (75 frames) after the pulse. All images of the first stack and the first ten images of the second stack were split into 8 × 8 pixel regions to compute a sinogram of every region. We identified the regions where the fluorescence signal was above a five photons count threshold as the foreground tiles. The user was able to change the threshold using a slider to manually adjust the detection if necessary. We calculated the Ca$^{2+}$ transient amplitude ($\Delta F/F$) on the identified foreground regions using the averaged baseline stack and the averaged two first frames of the response stack. The response size ratio was calculated as a ratio between the size of the area in which there was a minimum transient amplitude of 0.75 $\Delta F/F$ and the detected foreground area.

For STED imaging optimization, several objectives were considered. The autocorrelation amplitude of the actin periodic lattice was calculated from a line profile traced by the user on a STED image. More specifically, the amplitude was defined as the difference between the first maximum and the first minimum of the autocorrelation curve.

SNR and photobleaching were evaluated based on the detected foreground in STED and confocal images. Foreground was extracted by Otsu's method[58], which calculates the threshold between two modes (foreground and background) such that the variance in each mode is minimized. Let $\text{image}^p$ and $\overline{\text{image}}$ denote the $p$th percentile signal and the average signal, respectively, on an image. The SNR and photobleaching were estimated by:

$$\text{SNR} = \frac{\text{STED}_{\text{fg}}^{75} - \overline{\text{STED}_{\text{fg}}}}{\text{Confocal1}_{\text{fg}}^{75}} \tag{1}$$

and

$$\text{Photobleaching} = \frac{\overline{\text{Confocal1}_{\text{fg}}} - \overline{\text{Confocal2}_{\text{fg}}}}{\overline{\text{Confocal1}_{\text{fg}}}}, \tag{2}$$

where $\text{STED}_{\text{fg}}$, $\text{Confocal1}_{\text{fg}}$, and $\text{Confocal2}_{\text{fg}}$ respectively refer to the foreground of the STED image and confocal images acquired before and after. Finally, image quality rating was either manually performed by an expert user, or automatically performed by an FCN.

The FRC algorithm was implemented as shown in previous papers[18,19]. Two noise independent images are required to perform the analysis. Since the dataset for all experiments was already obtained at the moment of the publication of the methodology of FRC on STED images[18], only splitting of every image using a moving bin of $2 \times 2$ pixels was possible. We subsampled the original images (20 nm pixel size) into four independent images (40 nm pixel size). Each one of the pixels in the bin was attributed to their corresponding location in one of the four new images. We applied a Hann window ($\alpha = \beta = 0.5$), in both $x$ and $y$ directions, on the raw subsampled images to reduce the impact of high frequencies on the FRC curve as proposed in ref. [18] before the Fourier transform. We calculated the FRC on every possible combination of two from the set of four images (total of six combinations) and averaged the results to reduce the noise on the FRC curve. A moving average filter with a half-width of size 3 was also applied to smooth any remaining peaks. The subsampled images used to perform the FRC lacked frequency sampling due to their small size ($112 \times 112$ pixels). To lessen the problem of lack of information, we chose to use the 3-$\sigma$ threshold criterion to measure the resolution. The resolution of the image can be measured from the spatial frequency at which the FRC value falls below the threshold. Note that due to the large pixel size of the resampled images (40 nm), the resolution of STED images cannot be precisely calculated.

**Kernel TS for online optimization**. Classical formulation of mathematical optimization is to seek for the value $x$ in a domain $\mathcal{X}$ to maximize (or minimize) a real-valued target function $f : \mathcal{X} \mapsto \mathbb{R}$. Here, $f(x)$ corresponds to the expected objective value associated with the parametrization $x$ in the space of optimized parameters $\mathcal{X}$. The online optimization problem can be formulated as an episodic game, each episode corresponding to an algorithm selecting a parameter configuration from $\mathcal{X}$ and acquiring an image. This image leads to a noisy observation of the target function at the chosen parameterization. An algorithm evolving in this setting faces two challenges: it must estimate the target function from noisy observations acquired at different locations while selecting the right parameters to optimize the target function.

When the target function is known to be smooth but rigid parametric assumptions over this function are not realistic, kernel regression is able to estimate the target function from prior, noisy and not necessarily independent and identically distributed samples. More specifically, under multivariate Gaussian priors, kernel regression can provide the posterior distribution over a function given prior observations[59]. Whereas linear regression assumes that the target function is linear on its input space $\mathcal{X}$, kernel regression rather assumes that there exists a function $\phi$ mapping $\mathcal{X}$ to another space $\mathcal{K}$, such that the target function is described by a linear relation on $\mathcal{K}$. Formally, there exists a vector $\theta$ such that $f(x)$ corresponds to a inner product between $\phi(x)$ and $\theta$. (The linear setting corresponds to the specific case where $\phi(x) = x$.) The inner product between two mapped points $\phi(x)$ and $\phi(x')$ is given by the kernel $k(x, x')$. The choice of kernel controls the smoothness of the resulting regression model, thus encoding information about smoothness assumptions on $f$. In the experiments, we used an anisotropic Gaussian kernel (here defined for 1D):

$$k(x, x') = \exp\left(-\frac{(x - x')^2}{2\rho^2}\right) \tag{3}$$

with bandwidth $\rho_i = \ell_i D/3$ for parameter $1 \leq i \leq D$, with $\ell_i$ being the range of values for parameter $i$. We assume that the noise on observations of the target function is a zero-mean Gaussian with variance $\sigma^2$ and that $\theta$ follows a zero-mean multivariate Gaussian distribution with covariance matrix $\frac{\sigma^2}{\lambda}\mathbf{I}$, where $\mathbf{I}$ is the identity matrix, and $\lambda > 0$. The posterior distribution on $f$ after selecting $N$ parameters $x_1, \dots, x_N$ and observing $\mathbf{y}_N = (y_1, \dots, y_N)$ is then given by:

$$\mathbb{P}[f | x_1, \dots, x_N, y_1, \dots, y_N] \sim \mathcal{N}\left((f_{\lambda,N}(x))_{x \in \mathcal{X}}, \frac{\sigma^2}{\lambda}[k_{\lambda,N}(x, x')]_{x,x' \in \mathcal{X}}\right), \tag{4}$$

where

$$f_{\lambda,N}(x) = \mathbf{k}_N(x)^{\text{T}}(\mathbf{K}_N + \lambda\mathbf{I})^{-1}\mathbf{y}_N \tag{5}$$

and

$$k_{\lambda,N}(x, x') = \frac{\sigma^2}{\lambda}[k(x, x') - \mathbf{k}_N(x)^{\text{T}}(\mathbf{K}_N + \lambda\mathbf{I})^{-1}\mathbf{k}_N(x')] \tag{6}$$

respectively denote the predictive mean and (co-)variance (Supplementary Figure 1a), $\mathbf{k}_N(x)^{\text{T}} = (k(x_i, x))_{1 \leq i \leq N}$ and $\mathbf{K}_N = [k(x_i, x_j)]_{1 \leq i, j \leq N}$. Computing that posterior technically requires the knowledge of the noise variance $\sigma^2$, which is not realistic in practice. Fortunately, this can be circumvented by relying on empirical noise estimates based on lower and upper bounds on the noise variance, along with an upper bound on the norm of $\theta$[12]. All experiments used an upper bound of 5 on the norm of $\theta$ and a lower bound of 0.001 on the noise variance. An upper bound of 3 on the noise variance was considered for the maximal intensity objective (in glutamate uncaging), otherwise 0.3. The posterior distribution can be computed from the resulting noise estimate while preserving theoretical guarantees[12].

The parameters selection must trade off between two goals: improving the kernel regression model and aiming for the optimal parameters. Indeed, selecting the optimal parameters requires a model that is good enough to identify the optimum. This raises an exploration-exploitation trade-off typically addressed by bandits algorithms. Amongst bandits algorithms, TS[15] has gathered a lot of attention in the last years and has been the subject of many theoretical studies aiming at providing convergence guarantees through regret bounds analysis (e.g. ref. [60]). The general idea with TS is to select parameters according to their probability of being optimal. Concretely, this is achieved by acting upon samples from the posterior distribution given previous observations. Given that the posterior distribution is provided by kernel regression, the resulting Kernel TS[12] algorithm samples one function from the posterior distribution (conditioned on the history of previous observations) and selects the parameters optimizing that sampled function (Supplementary Figure 1b). An image is acquired with the selected parameters and the objective is computed on this image. The selected parameters and the observed objective value are then added to the history of previous observations, which will be used by kernel regression to provide the next posterior distribution.

**Kernel TS for online multi-objective optimization**. When optimizing toward multiple objectives, each objective is associated with a dedicated Kernel TS. Selecting the next parameter configuration requires that one function be sampled per objective, that is one function per posterior distribution (Supplementary Figure 6a). Each parameter configuration is now associated with a multi-objective sampled value constructed by joining the values of each sampled objective function for this parameter configuration (Supplementary Figure 6a). These multi-objective values form a cloud of options that are presented to the user such that he can pick his preferred trade-off among available options (Supplementary Figure 6b). An image is acquired with the parameters associated with the chosen option and the objective is computed on this image. The parameters and the observed objective value are then added to the history of previous observations, improving the knowledge on parameter space.

**Offline optimization with grid search**. The GS consists in discretizing the parameters space into equally spaced points and trying each point (one or more times) in order to estimate the objective value at that point. The resulting estimates are then used to select the best parameters configuration. This approach does not scale with the number of discretized parameters configurations. For example, if a single parameter is considered and discretized into ten parameter values, and five images are required to estimate the objective (e.g. autocorrelation amplitude), the GS will acquire $5 \times 10 = 50$ images. For three parameters, each discretized into ten parameter values, the GS would require $5 \times 10^3 = 5000$ images.

**Offline optimization with NSGA-II**. The Non-dominated Sorting Genetic Algorithm (NSGA-II)[25] is an MO optimization that works in an iterative fashion. The idea is to maintain a population of individuals (parameters configurations) that evolves over generations. On each generation, NSGA-II evaluates each individual from the current population and update the population for the next generation. Evaluating an individual (a candidate configuration) consists in acquiring images with the given configuration and averaging the values of objectives computed on each image. NSGA-II generates the next population with the aim to cover well the Pareto front of objectives values. After enough generations, the resulting population would typically contain individuals that lie somewhere on the front, therefore representing the available trade-offs between objectives. The optimal parameters configuration would then be selected by the user from that population. In our experiments, we performed ten generations of ten individuals per population, and acquired three images per evaluation. This approach required the acquisition of 300 images per optimization run. The experiments were conducted using the DEAP[61] NSGA-II implementation with bounded simulated binary crossover (crowding degree $\eta = 20$, probability of mating 0.9) and bounded polynomial

mutation (crowding degree $\eta = 20$, independent mutation probability (number of parameters)$^{-1}$).

**Offline optimization with random sampling**. The RS approach consists in selecting parameters configurations at random for characterizing the parameters space at a cheaper cost than with a GS.

**Automated quality rating**. The image quality rating is automated using a fully convolutional neural network (FCN)[62] with six convolutional layers. Starting from the input, convolutional layers have 32, 48, 48, 64, 64 and 64 $3 \times 3$ kernels. Each of these convolutional layers is followed by spatial batch normalization and an exponential linear unit activation. All layers but the last one are also followed by maxpooling with kernel $2 \times 2$, down sampling the image by a total factor of 32. The output is a linear combination of the last three layers. The weights for this combination are also learned as part of the training. This effectively allows the network to merge predictions from three different resolutions in its final decision, which correspond to down sampling factors of 32, 16 and 8 respectively. A sigmoid function is used to clip the values between 0 and 1. The global score for this image is then computed by averaging the score map, weighted by a foreground mask such that only the scores given to the foreground pixels are taken in consideration. It is also to be noted that the network being fully convolutional, it can accept arbitrarily size inputs. We defined the mask using simple foreground extraction on the STED image. Foreground was extracted using Otsu's method[58]. The mask was then resized to match the output of the network for a given input size and binarized (non-zero elements were set to one). The mask application is an essential part of the learning process, as it avoids the need to provide (and predict) ambiguous scores over areas without any content.

Datasets containing STED images of various cells and structures with different proteins and dyes were used for training and evaluating the models considered in the presented experiments (identifier: description). Each dataset was randomly divided into training and validation sets. The training set is the only one actually seen by the model in the training process, while the validation set is subsequently used to assert the true performance of the model on unseen data.

Actin: 2, 669 images (2, 135 training, 534 validation) of the actin cytoskeleton in fixed hippocampal neurons stained with phalloidin-STAR635.

Tubulin: 525 images (420 training, 105 validation) of the protein α-tubulin fixed hippocampal neurons stained with M-anti-Tubulin and GAM-Alexa 594 (104 images), GAM-STAR635P (103 images), GAM-STAR-RED (103 images), GAM-ATTO647 (105 images), and GAM-ALEXA633 (110 images).

αCaMKII: 757 (632 training, 125 validation) images of living neurons expressing GFP-αCaMKII.

PSD95: 494 images (395 training, 95 validation) of living neurons expressing PSD95-FingR-GFP.

LifeAct: 577 images (462 training, 116 validation) of living neurons expressing LifeAct-GFP.

Widefield: 400 (300 training, 100 validation) images of living neurons tagged with fluorescent proteins.

All these datasets, except for the Widefield dataset, were created from images obtained during user-driven optimization runs. The images were very small (around $4 \times 4$ μm), each acquired mostly in less than 5 s. For a completely new experiment, one could use tiling of already acquired larger images to rapidly generate an important number of images to train the network. For example, tilling of 100 $20 \times 20$ μm starting images would generate 2500 $4 \times 4$ μm training images. Already acquired images of various qualities obtained during routine widefield imaging of living neurons were used for the Widefield dataset. The images were $512 \times 512$ pixels.

A first FCN model was trained on the Actin dataset. Each input image was whitened by removing the mean and standard deviation computed over the Actin training set, such that the samples distribution exhibits a mean value of 0 and a standard deviation of 1. Considering the relatively small size of the training set (from a deep learning perspective), we used multiple data augmentation techniques to avert potential overfitting. Random arbitrary rotations were applied to the images, along with reflections and intensity alterations. ADAM[63] was used as SGD optimizer with a learning rate of 0.002 and parameters $\beta_1$ and $\beta_2$ set to 0.9 and 0.999, respectively. The learning rate was halved each 50 epochs to help convergence. Mini-batch size was set at 96. Root mean squared error (RMSE) was used as the loss function. A validation dataset (534 images distinct from the training set) was used to select the best network amongst all epochs (Supplementary Figure 39). We reached an overall RMSE of about 9% on the Actin validation set, although the error distribution vary widely depending on the actual quality of the image. For instance, for the very low scores, even an expert exhibits a high variance in its scoring, since the difference between 0 and 20% is not clear when both images are of poor quality anyway. The same argument can be applied for very high scores. The network uncertainty and variance is thus higher in these areas, which drives the RMSE (a metric sensitive to outliers) high. In the operational area—that is, in the area of interest for the optimization process—most of the model predictions incur less error w.r.t. the expert evaluation (Fig. 5c). Moreover, what matters the most is not the quality score in itself, but the ordering it creates among the captured samples, so the Kernel TS can use this information to suggest new parametrizations. Having a trained network then allows to learn using

much less data. We showcase this ability by testing the network on four smaller datasets, for which only a few hundreds of labeled images are provided (Supplementary Figure 40). To further assess the model generalization ability, we also conducted experiments where the FCN had access to only 96 training samples (Supplementary Figure 41). Finally, we provide results on an entirely different microscopy-related task, where the network has to learn to score images taken by a widefield microscope (Supplementary Figure 42).

**Automated preference articulation**. The preference articulation between objectives is automated using a shallow neural network (SNN)[64]. In opposition to the FCN, which is made of convolutional layers applying filters on their input, the SNN is made of fully connected layers applying dot products between their input and learned weights (Fig. 5a, d). The SNN contains two fully connected layers of ten artificial neurons wide. A rectifier linear unit activation follows each fully connected layer. The network takes an option as input, that is a vector of two elements (the image quality and the photobleaching caused by the imaging process) and outputs a score. By comparing scores given to different options from a cloud, we predict the choice of the expert as being the option with the highest score. This approach is independent of the cloud size and its distribution in the objective space.

We trained SNN models for articulating preferences given two and three objectives: maximizing quality, minimizing photobleaching and, optionally, minimizing total imaging time. An SNN model is trained using clouds of $N$ options. For each cloud $\mathcal{O}$, we built $N - 1$ pairs, each formed by the option $o_\star \in \mathcal{O}$, chosen by the user, and a non-chosen option $o_i \in \mathcal{O}, o_i \neq o_\star$. We trained the SNN in a ranking fashion with the margin ranking loss[65], which corresponds to a hinge loss with a tunable margin. For each pair $(o_\star, o_i)$, scores $s_\star$ and $s_i$ are predicted by the SNN using a forward pass on options $o_\star$ and $o_i$. The margin ranking loss is given by (recall that $\bigvee$ denotes max operator) $l_i = (s_i - s_\star - m) \vee 0$, where $m \geq 0$ is a margin. The network is thus penalized when the score of the chosen option $o_\star$ is below the score of the rejected option $o_i$ (within a certain margin $m$). This bears similarities with siamese networks[66,67]. The Adam[63] optimizer was used with a learning rate of 0.001, $\beta_1 = 0.9$ and $\beta_2 = 0.999$. We used batches of 128 pairs to estimate the gradient descent at each iteration.

The two-objective SNN was trained on 501 clouds obtained during fixed-cell optimization, each containing 728 options, using a margin $m = 0.1$. This left 62 clouds for validation and 62 clouds for testing. The three-objective SNN dataset contained 7620 clouds obtained during live-cell optimization, of different size: 5424 clouds with 800 options, 1254 with 1440, 699 with 960, 131 with 180, and 112 with 1920. We trained on 6096 clouds, leaving 762 clouds for validation and 762 clouds for testing, also using a margin $m = 0.1$. Training, validation, and test datasets were created by partitioning at the cloud level instead of pairs, to avoid observing instances of the same pair in the different datasets. Early stopping was performed using validation datasets when the network stopped improving during ten epochs (Supplementary Figure 43). Performance was assessed on test datasets, by computing the Euclidean distance between the option choice predicted by each SNN with the option chosen by the expert in the objective space. We observe that the median Euclidean distance corresponds to a choice that is very close to the expert choice in all objective dimensions (Fig. 5f, h).

**Bootstrap analysis**. Bootstrapping[68] consists in randomly resampling with replacement from an initial sample of a population in order to perform inference about the initial sample. These resamples can be used to compute statistics (such as confidence intervals) on the initial sample of the population. For example, consider a set of $N$ STED images. One can compute the average quality of these images, but not the distribution of the mean. This could be estimated by resampling $M$ times (with replacement) $N$ images from the initial set. The mean of each resample could be computed, along the distribution of those means.

The bootstrapping analysis from Fig. 4f considered $M = 1000$ samples with replacement from the response size ratios obtained using Kernel TS ($N = 640$) and an RS ($N = 480$). Each sample was used to compute the relative frequency of local response ratios at different thresholds. Confidence intervals were then computed at each threshold based on the 1000 relative frequencies.

**Statistical analysis of Ca$^{2+}$ uncaging optimization**. The results presented in Fig. 4g were analyzed using the non-parametric Kruskal−Wallis test (one-way ANOVA). For both the randomized and the optimization approaches, the null hypothesis (no change between images 1–20 and images 60–80) was tested for each response type (no response/low quality, widespread/low quality, local/low quality). More specifically, for each approach, the samples (values of relative frequency) computed over the first 20 images in each response type were compared with the samples computed for same response type over the last 20 images. For the randomized approach, each test was applied on $N = 6$ samples of relative frequency in images 1–20 and $N = 6$ samples of relative frequency in images 60–80. For the optimization approach, each test was applied on $N = 8$ samples of relative frequency in images 1–20 and $N = 8$ samples of relative frequency in images 60–80. This led to three results per approach, i.e. one result per response type.

**Replay experiment**. A replay experiment simulates an optimization run by using previously acquired images. In a replay experiment, each parameter configuration

is associated with a queue containing images previously acquired with this parameterization. A replay begins by shuffling the images inside each queue. Then, for a fixed number of iterations, the optimization approach selects parameters and obtains the next image from the associated queue. This simulates the imaging process. Under this simulation setting, two optimization techniques that select the same parameters obtain the same performance. This allows for a fairer comparison considering biological variations between samples.

For the replay experiments presented in Fig. 1, a dataset was built with 468 images that correspond to 39 images taken at 12 evenly spaced laser excitation power values from 0.2 to 21.1 µW in the back aperture of the objective. It contained consequently 39 images per queue, one replay trial consisted of 60 iterations and 100 trials were performed.

**Code availability**. Open source code for conducting MO optimization as described in this paper is available online: https://github.com/PDKlab/STED-Optimization. The provided software currently allows a user to perform out of the box optimization on an Abberior system equipped with Impsector 0.13, and instructions to define the proper platform-dependent interface functions otherwise. Optimization settings are configurable either through a command line interface or using a graphical user interface.

The FCN architecture for predicting the quality score of a STED image is available online: https://github.com/PDKlab/STEDQualityRatingFCN. The provided software currently provides a pretrained architecture ready to be deployed on a server and used within the optimization routine following proper instructions. Documentation for training new models is also provided. In order to minimize the setup time, we also make a public server available to serve image quality requests that can be used from a simple webpage. This webpage can be accessed at http://www.optim-nanoscopy.net.

The SNN architecture for learning a preference articulation function between several objectives is available online: https://github.com/PDKlab/STEDPreferenceSNN. The provided software currently provides a pretrained architecture ready to be deployed on a server and used within the optimization routine following proper instructions. Documentation for training new models is also provided.

## Data availability

All datasets used in the FCN experiments are available online: http://www.optim-nanoscopy.net/datasets/actin.zip, http://www.optim-nanoscopy.net/datasets/tubulin.zip, http://www.optim-nanoscopy.net/datasets/psd.zip, http://www.optim-nanoscopy.net/datasets/camkii.zip, http://www.optim-nanoscopy.net/datasets/lifeact.zip. All datasets used in the SNN experiments are available online: http://www.optim-nanoscopy.net/datasets/snn-2objs.zip and http://www.optim-nanoscopy.net/datasets/snn-3objs.zip. All additional raw data are available upon request.

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

## Acknowledgements
The authors thank Marie-Ève Paquet for making the CaMKII-SNAPf plasmid, Giuseppe Vicidomini and Giorgio Tortarolo for support with FRC analysis, Renaud Bernatchez for the implementation of a rating application, Simon Labrecque for the widefield imaging dataset, Francine Nault, Laurence Emond and Charleen Salesse for the neuronal cell culture. This work was supported by Natural Sciences and Engineering Research Council of Canada, the Canadian Institute of Health Research (PJT-153107), the Fonds de Recherche Nature et Technologie du Québec, and Mitacs.

## Author contributions
F.L.-C. and A.D. designed the experiments. F.L.-C., A.D. and M.-A.G. analyzed the data. A.D. developed and implemented the optimization algorithms and integrated them to the microscope. T.W., A.D. and A.B. developed the optimization routine for live-cell optimization. T.W., P.D.K. and F.L.-C. developed the multimodal imaging protocol. A.B. and F.L.-C. designed and implemented the photostimulation path on the STED microscope. T.W. and F.L.-C. performed STED experiments. M.-A.G., L.-E.R., A.D., and C.G. designed, characterized the performance, and implemented the neural networks. A.B. implemented the FRC analysis, developed the user interface and the online documentation. F.L.-C, A.D., T.W., C.G., and P.D.K. wrote the manuscript. F.L-C, A.D., C.G. and P.D.K. co-supervised the study. All authors reviewed the manuscript.

## Additional information

**Competing interests:** The authors declare no competing interests.

