## [Peer Review File · Nature Communications]

Reviewers' comments:

Reviewer #1 (Remarks to the Author):

The authors propose a fully-automated machine learning based system that conducts imaging parameter optimization simultaneously to the imaging task. They demonstrated the benefits of their approach in specific imaging experiments, by using image-based and expert-based quality indices. Overall, the potential impact is large, however I think the article is missing concrete "how-to" information in order to make it really useful for the community. Specific comments below:

C1:

"This strategy could be implemented easily on any existing microscope (without restriction to super-resolution) that is operated by a user, be it an expert or a non expert, conducting the optimization simultaneously as the imaging task. It would both improve the efficiency of the imaging process and standardize the obtained results."

I do not agree with this statement. It is absolutely not clear to me how someone could "easily" implement that method on their microscopy system. In order to make this study indeed useful for the community, those minimum requirements are needed: (i) open-source code with clear documentation and rapid online support and (ii) video/tutorial showing how to implement the method on an imaging platform.

C2:

Evaluation criteria might sometimes depend on the object feature, hence it is not clear to me how to compare different images with different object features. For example, in Fig1b, SNR for image #2 is lower than image #10, however it seems like there is also less bright object in the image, which would give a lower average SNR, independently from the intrinsic image SNR.

C3:

Figures are quite difficult to understand. Adding some context in the figure legend (what experiment, what do we see, etc.) would be helpful for the reader who might not be an expert in image optimization.

C4:

"Combined with deep learning methods to evaluate image quality and emulate user preferences, our optimization framework can automatically converge to well-performing imaging parameters without a user in the loop."

Is it really true? If a researcher was to start off with, let's say, imaging deep gray nuclei with electron microscopy. I'm assuming they would have to create a CNN model first, with expert rating and so on. And designing the CNN architecture for a given modality requires some design engineering which is not the expertise of every EM imager, therefore it is not clear how useful the proposed method would be for the everyday life a researchers, unless, as I said previously, a comprehensive how-to with open source code is provided.

C5:

This paper is about image quality optimization, but we rarely see images in the paper. Only metrics that can sometimes (and sometimes not) reflect the true assessment of an image quality. Simple figures showing images acquired with e.g., GS vs. proposed method would be useful for the reader.

Reviewer #2 (Remarks to the Author):

In this manuscript, Audrey Durand et al. describe a complete system to automatize the optimization of the parameters in advanced microscopy using deep learning. The authors present,

along the description, the experimental validation of this approach when applied to STED microscopy.

From my point of view, I agree with the authors that the multi-parametric optimization of a complex experimental system like the current Super Resolution (SR) techniques is an actual challenge. Nowadays, from a practical perspective, only microscopy experts are able to perform such improvement. Because of that, I agree with the authors that automatic approaches could lighten the access to optimized images to non-experts.

I have no doubts about the importance and motivation of the usage of deep learning in the problem exposed here. However, in general terms, I find an unbalanced effort to describe technically the results in comparison with the explanations and demonstrations of the usage and the benefits of using this approach. I really miss a description of the encountered limits of the technique, or at least a comment on where they could be. It is claimed that a non-expert user could obtain nice results using this approach. However, as example, there is no mention on a comparison between the results obtained with the effects of continuously acquiring images versus the acquisition of images with nice pre-settings configuration. In addition, I think that a general reader would expect a clear before-after optimization image and a comparison of experienced user versus automatic image. In fact, from my point of view the way of demonstrating the results is, in general, too technical and a non-specialized reader could not feel the real value of the approach. In the text below, I will provide the different points that justify my opinions from the results part. In the introduction, the authors describe that for optimizing the acquisition system parameters an exploratory phase has to be conducted. On the other hand, it is also exposed that the parameters can depend on plenty of factors and one of the most important is the sample itself. Then, it seems clear that this exploratory phase has to be performed anyway when training the Neural Network with a new type of samples.

Also in the introduction, the authors justify the usage of kernel TS to define the parameters space in comparison with the Grid Search. From my perspective, this is not a fair comparison with reality because the methodology to optimize a multi-parametric space can be performed by iteration algorithms like genetic algorithm, ant algorithm or others.

In the introduction, but this is a general impression for the whole results part, it is strongly emphasized the kernel TS algorithm but the usage of the deep learning algorithms (the CNN and the SNN) are not mentioned. In fact, I miss a global vision of the problem to solve and how it will be solved with each part to focalize and balance the attention of the reader in the critical points. The first results section demonstrates the benefits of using kernel TS versus GS using different merit functions in a one to one basis. From the text it is difficult to understand what is the aim of such comparison due to the technical level of the question exposed. In a first reading, for a non expert in algorithms and optimization problems, it could seem that only with this kernel TS the optimization procedure is fulfilled.

Emphasizing the previous comment, I think that the comparison with GS is unfair and also I would point that it would be nice to have an indication on the definition of the hyper-volume of possible solutions explored and how it is determined. The idea of this would be to show the difference between a full exploration of an unknown domain versus exploring a small volume of possible solutions. I really think that a priori knowledge can be used to avoid starting from scratch.

Following this line, the second section of results claims that GS is not suitable for multi-parametric search, which I completely agree. However, brute force testing is not the only solution for multi-parametric optimization and I would suggest to at least mention it but also I think that it would be great to see a demonstration of why kernel TS is the algorithm to use for parametric search.

In this second section, I had the impression that the discussion about the results was a bit confusing and a bit technical. From my point of view, the representation of results, with the present captions and explanations, are difficult to understand for non-experts in multidimensional analysis.

I would suggest fusing the second and third sections because I do not see the added differential value between them. The experiments are different but the conclusions are similar. Again, the use of RS as base line is a first approach but I would suggest to compare the kernel TS with other approaches.

As it is announced, in the title of the manuscript, the fourth section explains the usage of deep

learning architectures to automatize the optimization of parameters. It is explained how a CNN is used to evaluate the quality of the image as criterion for optimization. In this case, there is no reference or comparison on robustness of this approach in comparison with the previous ones. The same section describes also the usage of a SNN to mimic the decisions of a human expert in optimization of parameters in STED microscopy. I think that for a normal reader, the differences between CNN and SNN could be small and as they are described one after the other without any apparent contextualization, the reading of this section could become confusing.

Finally, in the fifth section of the results, the authors explain the architecture of the automatic machine learning approach for parameter optimization in STED microscopy. In this section, a single type of experiments is presented. This experiment demonstrate that the system really works in a practical way. In this part however, taking into account the amount of details performed in a partial section like the second and third, one could expect having more demonstrations of performance of the full system. In this case only a single type of sample and a single type of fluorophore is being used and also few parameters are optimized to improve the quality while it is claimed that the interest of using deep learning is to improve simultaneously many conditions. To summarize, I think that this manuscript is not able to demonstrate the capabilities of the approach to provide an automatic improvement of the parameters on STED microscopy and it is not able to demonstrate the reliability of the system in real experiments. I also think that the text is very technical and the results are presented in a very specialized way. Because all that, I cannot recommend the manuscript for publication in Nature communications but I would suggest that it could be submitted to a more specialized journal and I am sure that it would be a very nice contribution.

Reviewer #3 (Remarks to the Author):

The premise of their paper is the following: It is advantageous to have an algorithm which can optimise image acquisition settings. This is advantageous because it means potentially a skilled user is not required every time the system is optimised and also it introduces a degree of standardisation. To establish this method they use an optimisation algorithm called kernel Thompson Sampling and test a variety of loss functions which include those derived from signal processing principles as well as from neural networks trained to emulate human decision making in the same situation. In summary, I believe their contribution is valuable and novel, as it combines and proves several interesting methods for online optimisation. My main concern however is their choice of parameters to optimise, I think varying these parameters is not completely sensible for best practise imaging. Also I believe their methodology is somewhat distant from a practical solution as they are dependent on a human user to move the stage around between images and focus the sample.

Concerns

In Figure 2 and 3, these are not replay experiments with pre-acquired data, these images that are taken in sequence. Therefore is a human user required to move the stage and adjust the focus? This is not stated in the text as far as I could see. This obviously requires quite a lot of labour in the absence of an automated stage and his highly tedious. I think the authors should make clear that this is not ideal and would be substituted with an automated acquisition system for practical usage.

In their text they mention that their technique needed only 100 images compared to 3,840 potentially for the grid-search optimisation. For a STED experiment, even a 100 images is a large quantity to throw out and this would be pretty impractical for day-to-day usage. I find it very

surprising that they did not opt to perform their standardisation in at least part of a suitable standard such as beads, or even better Gattaquant nanorulers. This would show their system could be calibrated before being deployed on sensitive and perishable biological samples. Furthermore to train the CNN they required 2,477 STED images of fixed hippocampal neurons. This is really a lot of images. This is a proof of principle paper, but this is not practical for most users performing experiments.

From a perspective of good scientific practise, when performing a series of experiments the acquisition settings should be kept static over-time. However, that said, if you are using a system which is shifting in performance over-time (e.g. in laser output) then tuning the performance each time against a standard is important. This has been mentioned in the paper:

"The proposed approach could even rely on different quality assessment scores [26] for guiding the optimization, given that they could provide feedback in an interactive fashion.". The risk, as I see it, is that you may adapt the system too readily to compensate for differences between samples. Some of the variances may be as a result of your experimental treatment, rather than variances in your acquisition or sample preparation. If your staining is poor then no amount of optimisation in terms of the microscope can fix that and your optimisation will only obscure these problems. I recommend the authors make that clear in their discussion. Is the optimisation to be performed at the start of an experimental day, or is it really being tuned on every slide placed on the microscope?

Unfortunately for the authors the most significant aspect of a STED microscope which varies over-time is the alignment of the optics which generate the excitation psf and super-impose the depletion donut over the emitted fluorescence. Clearly their system at present does not allow automated control of the hardware elements which can affect this alignment. It would be fully advantageous to have a system which can optimise its alignment, whereas changing the laser power is quite subjective in terms of the experimental outcome. I think this point reduces the impact of the paper. The authors feel differently however: "Considering the high degree of variability in the expression level of the transfected protein, it is an important advantage to conduct the optimisation simultaneously to the imaging routine to ensure comparable results across experiments.". This is a contentious topic however and needs consideration on a case-by-case basis. In the case of their GCAMP response I think they are justified, but such an approach should be used with extreme caution.

Minor points

The paper is well written, but some of the sentences are little confusing due to repetition of nouns and lack of appropriate punctuation.

"This resulted in a higher sampling rate for the excitation power values (4.0 to 9.7 μ W) that are susceptible of maximizing the AA (Fig. 1d and Supplementary Fig. 4)."

"A replay experiment consists in simulating an experiment using previously acquired data in order to compare different approaches on the same data."

"This simulates the imaging process by giving access to similar images to all compared approaches given similar behaviours."

"For all fluorophores, the optimization process allowed to reach an image quality above 70%"

The paper is mostly well-presented. I found the figures with circles on quite confusing however e.g. Fig. 2b. Not sure if their maybe a better representation.

Reviewer #1 (Remarks to the Author):

The authors propose a fully-automated machine learning based system that conducts imaging parameter optimization simultaneously to the imaging task. They demonstrated the benefits of their approach in specific imaging experiments, by using image-based and expert-based quality indices. Overall, the potential impact is large, however I think the article is missing concrete "how-to" information in order to make it really useful for the community. Specific comments below:

"This strategy could be implemented easily on any existing microscope (without restriction to super-resolution) that is operated by a user, be it an expert or a non expert, conducting the optimization simultaneously as the imaging task. It would both improve the efficiency of the imaging process and standardize the obtained results."

I do not agree with this statement. It is absolutely not clear to me how someone could "easily" implement that method on their microscopy system. In order to make this study indeed useful for the community, those minimum requirements are needed: (i) open-source code with clear documentation and rapid online support and (ii) video/tutorial showing how to implement the method on an imaging platform.

We agree with the reviewer that the term “easily” was overstated, since the degree of difficulty to implement this method depends from the microscope on which it is implemented and the technical skills and expertise available. We now provide an open source code¹ with clear documentation and tutorial for both the online optimization and the deep network parts of the paper. Since parameter optimization will depend on the type of microscope, acquisition software, and objectives to achieve, some programming skills/expertise will be needed to implement our approach. However, we now clearly specify which modules require adaptation. Neural networks architecture are provided through a common software distribution interface (i.e., Docker²), allowing to reproduce the results of the paper and deploy these networks for usage by the online optimization system. Additionally, we now provide a graphical user interface that is adapted to an Abberior microscope but can be modified by the users of other microscopes. Online support will be provided via the GitHub repository when specific issues are faced. We modified the Discussion (p. 16) and the Methods (Data availability, deployment and reproducibility) in order to reference the available software.

Evaluation criteria might sometimes depend on the object feature, hence it is not clear to me how to compare different images with different object features. For example, in Fig. 1b, SNR for image #2 is lower

¹ Available at <https://github.com/PDKlab/STED-Optimization>, <https://github.com/PDKlab/STEDQualityRatingFCN> and <https://github.com/PDKlab/STEDPreferenceSNN>. The code is currently private – reviewers can access using the username *reviewer42* with password *bob123frigo*. All datasets and links to the Github repository can also be found on the web page : <http://www.optim-nanoscscopy.net>. Until submission, the page will stay private but can be accessed with the same password as for the Github repository.

² Docker (<https://docs.docker.com/>) provides a way to package an application with all its dependencies in an isolated container, allowing to run it safely and easily on another computer. It is available for Windows, Mac and Linux, and is widely used for distributing neural network models. The Github repositories currently provide Docker files that can be used to generate a docker image usable to retrain networks on new data, thus serving as a starting point for future research.

than image #10, however it seems like there is also less bright object in the image, which would give a lower average SNR, independently from the intrinsic image SNR.

The reviewer is correctly pointing out that strong intensity variations between structures and in different cell compartments (especially in neurons) strongly influence the evaluation criteria. This is the reason why classical approaches such as SNR or FRC evaluations sometime fail to evaluate the image quality (Supplementary Fig. 2&3). Nevertheless, image quality can be evaluated, independent of the object features, by the user. Therefore, we addressed this issue using the user-based quality rating approach. Furthermore, in our determination of the SNR, we performed a foreground extraction to avoid false SNR determination as mentioned by the reviewer (see Methods - Online analysis).

Figures are quite difficult to understand. Adding some context in the figure legend (what experiment, what do we see, etc.) would be helpful for the reader who might not be an expert in image optimization.

We have added relevant information in the figures and legends to improve their clarity and facilitate their interpretation.

"Combined with deep learning methods to evaluate image quality and emulate user preferences, our optimization framework can automatically converge to well-performing imaging parameters without a user in the loop."

Is it really true? If a researcher was to start off with, let's say, imaging deep gray nuclei with electron microscopy. I'm assuming they would have to create a CNN model first, with expert rating and so on. And designing the CNN architecture for a given modality requires some design engineering which is not the expertise of every EM imager, therefore it is not clear how useful the proposed method would be for the everyday life a researchers, unless, as I said previously, a comprehensive how-to with open source code is provided.

We agree with the reviewer that this is an important issue, which we addressed very seriously in this revised manuscript. We now propose an updated neural network architecture based on a fully convolutional network (FCN) (see Methods - Automated quality rating, Results Section 3, and Fig. 5a). This new architecture provides a much higher versatility since it is not limited anymore to a specific image size, format, or resolution. We also generate color-coded images as additional output that allow users to visually evaluate the rating given by the neural network (Fig. 5b & Supplementary Fig. 37, 38). We now provide stronger results showing that the proposed FCN model can be trained on images of various structures and with limited dataset size (see Results Section 3, Methods Automated quality rating, and Supplementary Fig. 37-40). Using this approach for images that are as different from STED microscopy images as electron microscopy images would probably require to retrain the network. Nevertheless, we now show that our FCN can be used to evaluate images that were acquired with a different microscope, hereby a widefield microscope (see Results Section 3, Methods - Automated quality rating, and Supplementary Fig. 42). Starting with a dataset of 400 widefield images of variable quality, we could retrain the FCN from scratch or use our fine-tuning approach based on our Actin dataset to generate reliable quality ratings of widefield images (Supplementary Fig. 42). This demonstrate that our network can be successfully retrained and is not limited to the quality rating of STED images.

The provided repository³ contains proper documentation to retrain a model on new data. We modified the Discussion p. 16 and Methods (Data availability, deployment and reproducibility) in order to reference the available software. We agree that training a neural network requires some technical and programming skills. We will provide support for researcher that would like to implement this approach with our pre-established architecture on their system. Our software should be adaptable by investigators, according to their needs.

This paper is about image quality optimization, but we rarely see images in the paper. Only metrics that can sometimes (and sometimes not) reflect the true assessment of an image quality. Simple figures showing images acquired with e.g. GS vs. proposed method would be useful for the reader.

Examples of high and low quality images for diverse structures examined are now available (Supplementary Fig. 14, 16, 17, 20, 23, 28, 31, 33, 44).

We also included example images taken with several methods presented in this paper: single objective Kernel TS, multi-objective Kernel TS, NSGA-II, and fully automated imaging platform (Fig. 1, 2, 3, 4, 5 & 6, Supplementary Figures 9, 14, 16, 17, 20, 23, 28, 31, 33, 44).

Reviewer #2 (Remarks to the Author):

In this manuscript, Audrey Durand et al. describe a complete system to automatize the optimization of the parameters in advanced microscopy using deep learning. The authors present, along the description, the experimental validation of this approach when applied to STED microscopy.

From my point of view, I agree with the authors that the multi-parametric optimization of a complex experimental system like the current Super Resolution (SR) techniques is an actual challenge. Nowadays, from a practical perspective, only microscopy experts are able to perform such improvement. Because of that, I agree with the authors that automatic approaches could lighten the access to optimized images to non-experts.

I have no doubts about the importance and motivation of the usage of deep learning in the problem exposed here. However, in general terms, I find an unbalanced effort to describe technically the results in comparison with the explanations and demonstrations of the usage and the benefits of using this approach.

We added more details about the usage and the benefits of Kernel TS in diverse contexts throughout the results section and discussion. See for example p. 4 (optimization with different fluorophores), p. 7 and new Figure 3 (live-cell imaging in various cell type), p. 11 (training of fully convolutional neural networks with datasets containing a small number of images, of different structures and acquired with another type of microscope), p.12 (fully automated optimization of different types of samples). Images acquired along the fully-automated optimization sequences have also been added to Figure 6 and to Supplementary Fig. 44 and 46.

I really miss a description of the encountered limits of the technique, or at least a comment on where they could be.

³ <https://github.com/PDKlab/STEDQualityRatingFCN>

The main limitation of the presented approach is the need to retrain the quality rating network for different type of subjects (proteins, structures) and the preference articulation network for different objectives trade-offs. We addressed this by providing open source code (links given in previous responses) with clear documentation. These repositories contain interfaces for using and training both neural networks (the FCN and the SNN), with the complete documentation detailing the installation process, the usage, and the training procedure. We modified the Discussion (p. 16) and Methods (Data availability, deployment and reproducibility) in order to reference the available software. As for the online optimization algorithm based on Kernel TS, it suffers from the common limitations of kernel regression approaches: that is it does not scale well with the number of inputs (here images) and prediction points (here evaluated parameters configurations). This is usually not a problem in the online setting where the number of samples is already limited, therefore so is the search space. However, given that we wished to conduct optimization sequences over very large number of images (more than 10 000 images) in a larger parameter space (more than 4 dimensions), one could rely on recent works (Pleiss 2018) to circumvent this issue. One should also note that the number of images required for finding a (near-)optimal solution scales with the number of effective dimensions in the search space, as the sharing of information decreases as the number of dimensions increases (Srinivas 2010, Valko 2013). This, however, is not a limitation of the proposed approach, but an inherent limitation of minimal space coverage. This has been added to the Discussion (p. 16).

It is claimed that a non-expert user could obtain nice results using this approach. However, as example, there is no mention on a comparison between the results obtained with the effects of continuously acquiring images versus the acquisition of images with nice pre-settings configuration. In addition, I think that a general reader would expect a clear before-after optimization image and a comparison of experienced user versus automatic image. In fact, from my point of view the way of demonstrating the results is, in general, too technical and a non-specialized reader could not feel the real value of the approach.

We added before and after optimization images for various experiments in the supplementary material to help readers appreciate the benefits of the proposed approach (Supplementary Fig. 9, 11, 14, 16, 17, 20, 23, 28, 31, 32, 33, 44). We also compared the online optimization process using Kernel TS in a larger parameters space with the optimization performed by an expert (it is now mentioned in the Results section 2 (p. 6)). The results (Supplementary Fig. 11) show that Kernel TS was able to converge to the same region of parameters as the expert, even though Kernel TS had to search a much wider space. For a new type of sample that is not known (even for an expert user) Kernel TS can be employed to search efficiently a large parameter space, while attempting to acquire as many good images as possible (exploration-exploitation trade-off). We also added example images acquired at the end of the Kernel TS optimization sequence and by the expert user (Supplementary Figure 11c).

Although it may be tempting to use the prior knowledge of the user (expert or not) to tighten the search space, one must be careful that the range of parameters actually contains an *optimal* configuration. As we showed for tubulin tagged with three different fluorophores (Alexa 633, ATTO 647 and STAR RED) (Fig. 2b), with different staining concentration for the actin cytoskeleton (Supplementary Fig. 32) or with diverse proteins tagged with the green fluorescent protein (GFP) (Fig. 3 & Supplementary Fig. 15-31), the *optimal* configuration may depend heavily on the experimental conditions, the fluorophores and the observed structures. Given these variables, pre-optimization is often not optimal, at least in comparison to Kernel TS-based online optimization. We now discuss in the text the limitations of pre-optimization on p. 7-8.

In the text below, I will provide the different points that justify my opinions from the results part.

In the introduction, the authors describe that for optimizing the acquisition system parameters an exploratory phase has to be conducted. On the other hand, it is also exposed that the parameters can depend on plenty of factors and one of the most important is the sample itself. Then, it seems clear that this exploratory phase has to be performed anyway when training the Neural Network with a new type of samples.

We have clarified the Introduction and Results sections 1&2 (p. 3-4) to remove confusion regarding online versus offline optimization and the pure exploration phase induced by the latter. Essentially, our paper proposes an online optimization system in order to avoid the aforementioned pure exploration phase. This optimization can be performed either with a user in the loop or automatically using pre-trained neural networks. For fully-automated optimization using the quality objective, one should test prior to an optimization run if the FCN is able to perform well with the structure of interest using a small test dataset of the structure to image. Note that the FCN is trained once for a family of structures, not on every parameter optimization task, nor per sample. This has been added to the Results section 3 (p. 11), the Methods (Automated quality rating) and to the Discussion (p. 16).

We added several experiments (see Results section 3) to characterize the situation where an FCN model must be trained on new data, and show that fully-automated optimization can be conducted using FCNs fine-tuned on smaller datasets (Supplementary Fig. 39-41). These images need not to be acquired in a pure exploration phase: they can be generated using online optimization runs with a user in the loop. The fully-automated functionality therefore becomes naturally available once after a few manual runs. We now refer to the training of the FCN with datasets of different sizes and various structures in the text (p. 11) and in the Methods (Automated quality rating - Datasets and Models).

Also in the introduction, the authors justify the usage of kernel TS to define the parameters space in comparison with the Grid Search. From my perspective, this is not a fair comparison with reality because the methodology to optimize a multi-parametric space can be performed by iteration algorithms like genetic algorithm, ant algorithm or others.

In our paper, the goal is to conduct multi-objective optimization simultaneously as the imaging task. In other words, we try to maximize the number of good images during the optimization phase. This is different from conducting optimization in a first phase (also known as pure exploration), followed by the phase of imaging (also known as exploitation). This has been clarified in the Introduction (p. 2). Genetic algorithms (like NSGA-II) are pure exploration algorithms. Pure exploration algorithms do not try to obtain good images during the optimization: they are designed to heavily search the space of solutions. Moreover, in the multi-objective setting, they search for any solution that lies on the Pareto front (the front built by non-dominated trade-offs between the objectives)— the idea being that the trade-off would be performed in the next exploitation phase. Therefore, they require a large quantity of images to identify all solutions on the front, but the user will typically care about a single region of the front (which is not known in advance).

To demonstrate this point and address the issue raised by the reviewer, we performed experiments using the well-known genetic algorithm NSGA-II for multi-objective optimization (Results section 2, p. 4 and

Methods - Offline optimization). Results on a simulated model built from real data show that this algorithm is indeed capable of identifying the Pareto front, but not while maximizing good images acquired during the search (Supplementary Fig. 7). We confirmed our simulation results on experimental data using NSGA-II as a benchmark and Kernel TS. When using NSGA-II we could identify solutions on the Pareto front (trade-offs between objectives) but with a larger number of images. In our experiments using GATTA-STED Nanorulers, Kernel TS generated 69 out of 100 images with high image quality (score above 60%) in comparison with only 37 for NSGA-II. When imaging the actin cytoskeleton tagged with Phalloidin STAR635, Kernel TS generated 46 high quality images out 100 images, while it was only 27 for NSGA-II. Even when pursuing the NSGA-II optimization up to 330 images for GATTA-STED Nanorulers, only 129 (39%) high quality images were obtained, and for the actin cytoskeleton only 100 (33%) high quality images were obtained out of 300 images. The corresponding additional results are shown in Supplementary Fig. 7&8.

In the introduction, but this is a general impression for the whole results part, it is strongly emphasized the kernel TS algorithm but the usage of the deep learning algorithms (the CNN and the SNN) are not mentioned. In fact, I miss a global vision of the problem to solve and how it will be solved with each part to focalize and balance the attention of the reader in the critical points.

To address this important issue, we made considerable changes to the introduction and the sections about the neural networks and fully automated optimization.

The first results section demonstrates the benefits of using kernel TS versus GS using different merit functions in a one to one basis. From the text it is difficult to understand what is the aim of such comparison due to the technical level of the question exposed. In a first reading, for a non expert in algorithms and optimization problems, it could seem that only with this kernel TS the optimization procedure is fulfilled.

The first and second results section has been amended to better highlight the fact that online optimization (Kernel TS) pursues a different goal than offline optimization (e.g. grid search (GS) or NSGA-II).

One reason to compare Kernel TS with GS is that the latter is commonly used in biology.

Kernel TS aims at maximizing the performance (in this case acquire good images) simultaneously as conducting the optimization, whereas offline optimization methods (such as GS) aim at maximizing the performance of the recommended parameterization resulting from a pure exploration phase. GS is an example that is commonly used for one-parameter single-objective optimization. In this setting, other approaches (e.g. Kernel UCB, GP-UCB) could also provide similar results as Kernel TS, as they are designed for online single-objective optimization. However, in the multi-objective setting, Kernel TS is used to generate options that are shown to the user, who is in charge of making the proper trade-off between objectives. In order for the user to make informed decisions, the presented options must be meaningful. Kernel TS is an intuitive choice as generated options are informative about the expected objective function. Kernel UCB/GP-UCB would generate deterministic options that would always correspond to the upper-bound on functions. Other GP-based algorithms (e.g. EI) compute values that are well suited for optimization, but not for a user choice, as these values mean nothing to the user. This has been specified in Results section 2 (p. 4).

Emphasizing the previous comment, I think that the comparison with GS is unfair and also I would point that it would be nice to have an indication on the definition of the hyper-volume of possible solutions explored and how it is determined. The idea of this would be to show the difference between a full exploration of an unknown domain versus exploring a small volume of possible solutions. I really think that a priori knowledge can be used to avoid starting from scratch.

The hyper-volume of possible solution is defined by the user (or could be a standardized hyper-volume set by an expert on a given microscope). As shown in Fig 2, where Tubulin tagged with three dyes (STAR RED, ATTO647 and Alexa633) was imaged, the optimal imaging range can be very different between fluorophores. Therefore, if little knowledge about the photophysical properties of a new dye is provided, the hyper-volume should be large to avoid missing optimal measuring ranges. We now show that prior knowledge can be used to reduce the hyper-volume for very similar samples, but that this approach can be risky (Supplementary Fig. 32). We used two different concentrations of phalloidin-STAR635 to mark the actin cytoskeleton in neurons. We first performed for each sample an optimization run using Kernel TS. We then reduced the hyper-volume of parameters to the most sampled regions in both optimization runs. With those settings, we performed 50 further images on the lower concentration sample. The quality of the obtained images is strongly decreased when using the reduced hyper-volume from the high concentration sample (Supplementary Fig. 32d). This can be explained by the fact that the smaller hyper-volume for the high concentration sample is outside the optimal zone for the low concentration sample. The reduction of the hyper-volume is indeed possible when using prior knowledge acquired with the same sample (or very similar) and lead rapidly to high quality images (Supplementary Fig. 32, 33). We clarify this issue in Results section 2 (p. 4).

We also investigated the use of prior knowledge in the context of live-cell imaging of GFP tagged samples. We compared the results obtained with full optimization sequences performed on various proteins (CaMKII-GFP, LifeAct-GFP, Membrane-GFP) and cell types (Neurons, HEK and PC12) or with shorter optimization sequences using prior knowledge of two reference samples (GATTAQUANT Nanobeads OG488 and PSD95-FingR-GFP in neurons). We show in several supplementary figures (Supplementary Fig.18, 19, 21, 22, 24, 25, 26, 27, 29, 30) that previous knowledge can not robustly improve the results and we now refer to this in the manuscript (p. 7-8).

Following this line, the second section of results claims that GS is not suitable for multi-parametric search, which I completely agree. However, brute force testing is not the only solution for multi-parametric optimization and I would suggest to at least mention it but also I think that it would be great to see a demonstration of why kernel TS is the algorithm to use for parametric search.

Kernel TS is not used alone (as in single objective): here Kernel TS is used to produce estimates that are presented to an expert (user or SNN). We are not sure if the reviewer is asking 1) whether TS is the best choice to use with kernel regression (e.g. why not EI or UCB) or 2) whether an approach based on kernel regression is the best way to go (e.g. use a genetic algorithm to search the space of parameters). In the case of 1), we addressed it above; in the case of 2), we provide new results with NSGA-II, noting that it does not apply to online optimization (Supplementary Fig. 7, 8, and Results section 2, p. 4). We thus believe indeed that Kernel TS is a proper algorithm to use in the context presented in this paper.

In this second section, I had the impression that the discussion about the results was a bit confusing and a bit technical. From my point of view, the representation of results, with the present captions and explanations, are difficult to understand for non-experts in multidimensional analysis.

I would suggest fusing the second and third sections because I do not see the added differential value between them. The experiments are different but the conclusions are similar. Again, the use of RS as base line is a first approach but I would suggest to compare the kernel TS with other approaches.

We clarified the captions and explanations in this section. We fused both sections, as suggested, but kept both figures and added a new figure (Fig. 3) to highlight the applicability of our approach to real-life experiments. We think that the use of Kernel TS optimization for a multi-modal experiment is very important since it 1) shows that Kernel TS is not limited to super-resolution microscopy and 2) is a good example of “real life” application when applied to a broadly used method in neuroscience : glutamate uncaging.

A fair comparison of Kernel TS with genetic algorithms, that require a very large number of images (more than 100), is a difficult task. It is unsuitable for live-cell imaging or multi-modal experiments (Fig. 2, 3 and 4) due to the time limitation when imaging living cells on the microscope. Instead, we compared NSGA-II and Kernel TS for fixed cell imaging of phalloidin samples and GATTAQUANT Nanorulers. ¶

If the imaging of a large number of regions is possible, NSGA-II provides a reliable evaluation of the Pareto front. It is performed as a prior exploration phase to the real imaging session. The obtained results can be further used to determine the best parameter combination to use for the imaging session. It is a good method for so-called offline optimization. With Kernel TS, we aim to avoid a prior exploration phase and propose an online optimization approach. Comparison with RS in the case of multi-modal experiment is related to the very limited number of images for a given neuron (maximum 80) that would be incompatible with NSGA-II. We modified the text in accordance to this comment in the Results section 2 (p. 4). We added Supplementary Fig. 8 to depict the performances of Kernel TS and NSGA-II algorithms on Phalloidin STAR635 and GATTA-STED Nanorulers 70 nm ATTO647N.

As it is announced, in the title of the manuscript, the fourth section explains the usage of deep learning architectures to automatize the optimization of parameters. It is explained how a CNN is used to evaluate the quality of the image as criterion for optimization. In this case, there is no reference or comparison on robustness of this approach in comparison with the previous ones. The same section describes also the usage of a SNN to mimic the decisions of a human expert in optimization of parameters in STED microscopy. I think that for a normal reader, the differences between CNN and SNN could be small and as they are described one after the other without any apparent contextualization, the reading of this section could become confusing.

In order to compare the success of the optimization conducted with automated quality rating using the FCN versus the user-rated quality optimization, we now provide explicit figures depicting the error made by all studied FCN models on their validation set (Fig. 5c and Supplementary Fig. 40-42) along with images (Fig. 6e and Supplementary Fig. 44) obtained during fully-automated optimization. We also provide score maps obtained with all studied FCN models on the validation sets (Fig. 5b and Supplementary Fig. 37, 38).

We clarified the distinction between a FCN and a SNN in Results section 3 (p.11).

Finally, in the fifth section of the results, the authors explain the architecture of the automatic machine learning approach for parameter optimization in STED microscopy. In this section, a single type of experiments is presented. This experiment demonstrate that the system really works in a practical way. In this part, however, taking into account the amount of details performed in a partial section like the second and third, one could expect having more demonstrations of performance of the full system. In this case only a single type of sample and a single type of fluorophore is being used and also few parameters are optimized to improve the quality while it is claimed that the interest of using deep learning is to improve simultaneously many conditions.

We addressed this issue by adding several experimental results performed with the fully-automated system using a FCN trained on the Actin dataset (see Methods) for imaging different proteins (Actin-STAR635, Tubulin-STAR635P, PSD95-STAR635P, Bassoon-STAR635P, GluN2B-STAR635P) (Fig. 6 and Supplementary Fig. 45). Note that the FCN trained on the Actin dataset had never seen these proteins before. This shows that the proposed FCN can generalize well and would not necessarily need to be retrained for a new protein or structure. We also present new results for the image quality rating using a FCN model trained on the Actin-dataset (Phalloidin-STAR635 in fixed neurons) and fine-tuned on various other datasets (PSD-dataset (PSD95-FingR-GFP in living neurons), CaMKII-dataset (GFP- α CaMKII in living neurons), Tubulin-dataset (Mouse-anti-tubulin with various fluorophores in fixed neurons), LifeAct-dataset (LifeAct-GFP in living neurons), Widefield-dataset (widefield images of living neurons expressing various proteins tagged with fluorescent proteins)) (Results section 3, p. 11, and Supplementary Fig. 40-42). We also added results obtained during fully-automated imaging on live cells using those fine-tuned FCN (Fig. 6 and Supplementary Fig. 44, 46), showing that an FCN model can be further specialized toward specific structures or image resolutions. Additionally, we trained our SNN in the context of two and three objective optimization to show that this approach can be adapted to a new type of optimization task (Fig. 5d-h and Supplementary Fig. 43). These additional experiments demonstrate the wide applicability of our system and its performance.

To summarize, I think that this manuscript is not able to demonstrate the capabilities of the approach to provide an automatic improvement of the parameters on STED microscopy and it is not able to demonstrate the reliability of the system in real experiments.

To fully demonstrate the capabilities of the approach, we provide major revisions in this new version, including a large set of new experimental results. Using various proteins, we show that the proposed FCN model for rating image quality can be retrained on much smaller datasets and still remains a valid target to drive optimization. We also show the flexibility of the SNN model for preference articulation, that can be taught to trade off between different objectives and different number of objectives. Finally, we added many fully-automated experiments where we optimized the imaging of various proteins in fixed and living cells (Actin, Tubulin, PSD95, CaMKII, Bassoon, GluN2B, LifeAct) using different fluorescent markers (STAR635, STAR635P, GFP). We also added several examples of images acquired for the experiments performed throughout the paper (see Fig. 3,4,6 and Supplementary Fig. 14, 9, 33, 16, 17, 20, 23, 28, 31, 44). Images acquired during fully-automated runs now clearly showcase the capability of the proposed system for efficiently optimizing imaging parameters without the need of an expert user.

I also think that the text is very technical and the results are presented in a very specialized way. Because all that, I cannot recommend the manuscript for publication in Nature communications but I would suggest that it could be submitted to a more specialized journal and I am sure that it would be a very nice contribution.

The added experiments and results in this revised manuscript should broaden significantly the interest to the Nature Communications readership. Moreover, we now provide the optimization framework and the neural networks components in an online repository with proper documentation for deploying the system, retraining neural networks, building datasets, and transferring this approach to other platforms. This is now detailed in the Discussion (p. 16) and Methods (Data availability, deployment and reproducibility).

Reviewer #3 (Remarks to the Author):

The premise of their paper is the following: It is advantageous to have an algorithm which can optimise image acquisition settings. This is advantageous because it means potentially a skilled user is not required every time the system is optimised and also it introduces a degree of standardisation. To establish this method they use an optimisation algorithm called kernel Thompson Sampling and test a variety of loss functions which include those derived from signal processing principles as well as from neural networks trained to emulate human decision making in the same situation. In summary, I believe their contribution is valuable and novel, as it combines and proves several interesting methods for online optimisation. My main concern however is their choice of parameters to optimise, I think varying these parameters is not completely sensible for best practise imaging.

The parameters we chose to demonstrate the applicability and the robustness of online Kernel TS optimization are among the most frequently modified parameters in day to day practice for STED microscopy. To obtain good results with STED microscopy, these imaging parameters (1. excitation laser power, 2. depletion laser power, and 3. pixel dwell time) must be adjusted for each imaged structure and staining. We decided that varying these same parameters throughout the demonstration was important for the readers to compare results across experiments. We also show that we can apply Kernel TS optimization to a new context (glutamate uncaging) and introduce new parameters (uncaging laser power and uncaging distance from dendritic spine) (Fig. 4 and Results section 2, p. 8).

Also I believe their methodology is somewhat distant from a practical solution as they are dependent on a human user to move the stage around between images and focus the sample.

The objective of this approach was not to design a robotic microscope, but to design an automated optimization of the imaging parameters of a typical complex microscope that scientists are using. It was not our intention to substitute human intervention in operating the microscope, as sample and region of interest selection remains a key intervention for observation-based microscopy of relevant biological structures.

Our STED microscope was equipped with an automated stage and auto-focus unit. The imaging sequence was the following: 1) with a Python routine integrated in our optimization software, the user defined around 30 regions of interest in a given imaging field; 2) those regions were then sequentially scanned; 3) for each

region a new combination of parameters was used; and 4) new options were shown to the user (or SNN) after each image to choose the next imaging parameters. This imaging sequence was performed without the need for the user to move the stage or re-adjust the focus. We modified the text to clarify this procedure in the methods section (STED imaging, live-cell imaging, and glutamate uncaging). This imaging routine is now available and clearly documented on our GitHub repository (Methods section Data availability deployment and reproducibility).

In Figure 2 and 3, these are not replay experiments with pre-acquired data, these images that are taken in sequence. Therefore is a human user required to move the stage and adjust the focus? This is not stated in the text as far as I could see. This obviously requires quite a lot of labour in the absence of an automated stage and his highly tedious. I think the authors should make clear that this is not ideal and would be substituted with an automated acquisition system for practical usage.

A user was in the loop to rate the images and choose the regions to image. This was done using an overview image of the whole scanning range and pointing the interesting regions to image with a pointer controlled with the computer mouse. No refocusing was needed since the microscope is equipped with an autofocus system and the automated stage was controlled with the Python optimization script, now provided as an open source code (see Discussion, p. 16, and Methods - Data availability, deployment and reproducibility). Such features may differ between microscopes but are indeed extremely helpful to speed up the imaging process.

Around 20-30 regions of about 4 x 4 μm were chosen along the neurons and automatically imaged in sequence. This automated scanning of multiple regions was implemented as a part of the whole optimization system. One of the key conclusion of our study is that these online optimization steps endup accelerating the achievement of the main goal which is to acquire good images with the microscope. One should emphasize that online optimization allows to simultaneously optimize imaging parameters and maximize the number of good images. We updated the Methods section (STED imaging, live-cell imaging, and glutamate uncaging) with a detailed description of the imaging sequence and the use of the motorized stage and auto-focus units .

In their text they mention that their technique needed only 100 images compared to 3,840 potentially for the grid-search optimisation. For a STED experiment, even a 100 images is a large quantity to throw out and this would be pretty impractical for day-to-day usage.

The 100 images are very small (224 x 224 pixels, $\sim 4 \times 4 \mu\text{m}$) and their acquisition takes on average less than 5s each. We added this information to the Methods section (STED imaging, live-cell imaging, and glutamate uncaging). In the fully-automated setup, 100 images are taken in less than 30 minutes. We show by comparing Kernel TS with grid search (GS) and the genetic algorithm NSGA II (Fig. 1, Supplementary Fig. 8, and Results section 2, p. 4) that acquiring STED images without pre-optimized parameters would generate a larger number of low quality images. We also now clarify the main advantage of using online optimization, which is that the optimization is performed during the imaging process (Introduction and Results sections 1&2, p. 3-4). The images that are obtained during online optimization process and that satisfy objective trade-offs can further be used for analysis.

I find it very surprising that they did not opt to perform their standardisation in at least part of a suitable standard such as beads, or even better Gattaquant nanorulers. This would show their system could be calibrated before being deployed on sensitive and perishable biological samples.

We had not performed those standardisation experiments since we experienced that beads are not suitable to calibrate the STED parameters for very specific imaging tasks such as live-cell imaging, especially with fluorescent proteins. This pre-calibration would result in misleading optimal parameters. But as the reviewer mentioned, it is important to address this question to better explain the necessity of online parameter optimisation. We thus performed various experiments with QUATTAQUANT nanobeads (Oregon Green 488) and GATTA-STED nanorulers (ATTO647). We now show (Supplementary Fig. 15-31) that the obtained parameters and results for GATTAQUANT nanobeads are very different than the one needed for live-cell imaging of GFP. We also refer to the optimization using pre-setting obtained from the optimization with GATTAQUANT Nanobeads in the Results section 2 (p. 7-8). Due to the high photostability of the nanobeads, a higher excitation and STED power were used to obtain two high quality super-resolution images, while minimizing the photobleaching and the imaging time. However, when applied to GFP-tagged structures, the use of previous knowledge from the nanobeads led to very high photobleaching of GFP for most of the observed structures (Supplementary Fig. 18, 19, 21, 22, 24, 25, 26, 27, 29, 30).

Additionally, we used the GATTA-STED Nanorulers when comparing Kernel TS and NSGA II optimization sequences (Supplementary Fig. 8a-b). The results further confirm those obtained for the imaging of the actin cytoskeleton in neurons (Supplementary Fig. 8c-d).

Furthermore to train the CNN they required 2,477 STED images of fixed hippocampal neurons. This is really a lot of images. This is a proof of principle paper, but this is not practical for most users performing experiments.

The dataset was created with the images obtained during the optimization process. Those were very small images (~ 4 x 4 um) that were acquired mostly in less than 5 s per image. For a completely new experiment, one could for example use tiling of already acquired large images to generate rapidly a large number of images to train the network (for example, tiling of 20 x 20 um images would generate the same number of small images with only 100 starting images). We added this information to the Methods section (Automated quality rating - Datasets).

To demonstrate the capacity of our approach to perform well with a more limited set of images, we conducted experiments using FCN models fine-tuned on smaller datasets of different proteins (tubulin, PSD95, CaMKII, LifeAct). We observed good performance of these models for predicting quality ratings by an expert (Supplementary Fig. 40). We then characterized the performance of the proposed FCN model fine-tuned on less than 100 images (Supplementary Fig. 41). All datasets (now detailed in Methods - Automated quality rating) were generated during user-driven optimization runs and are now available online. The Methods (Data availability, deployment and Reproducibility) now provide detailed documentation and open source codes for users to have the possibility to train the FCN with their datasets.

From a perspective of good scientific practice, when performing a series of experiments the acquisition settings should be kept static over-time. However, that said, if you are using a system which is shifting in performance over-time (e.g. in laser output) then tuning the performance each time against a standard is important. This has been mentioned in the paper:

“The proposed approach could even rely on different quality assessment scores [26] for guiding the optimization, given that they could provide feedback in an interactive fashion.”. The risk, as I see it, is that you may adapt the system too readily to compensate for differences between samples. Some of the variances may be as a result of your experimental treatment, rather than variances in your acquisition or sample preparation. If your staining is poor then no amount of optimisation in terms of the microscope can fix that and your optimisation will only obscure these problems. I recommend the authors make that clear in their discussion. Is the optimisation to be performed at the start of an experimental day, or is it really being tuned on every slide placed on the microscope?

The reviewer makes a good point that we now address in the discussion. One could reverse the argument by considering that a low quality (or low intensity) staining combined with non-suitable acquisition parameters could lead to a wrong interpretation of the biological information. Indeed, if a user would optimize the imaging parameters on a given sample and always reuse them without adapting, wrongly chosen parameters could give the impression that a feature does not exist just because it is not detected.

As we now show (Supplementary Fig. 32), inadequate parameter adjustment can indeed induce misleading interpretation of the biological structures that can be recognized in the acquired images. Indeed, we compared the actin rings on axons that can be identified in different phalloidin staining on fixed hippocampal neurons (Supplementary Fig. 32 and Results section 2, p. 7-8). We first performed independent Kernel TS optimization sequences with 100 images for each sample (low concentration sample (phalloidin concentration of 1:1000) and high concentration sample (concentration of 1:100)). We then reduced the hyper-volume of parameters to the combination range that was mostly sampled during the full optimization sequence for each sample. We performed two optimization sequences of 50 images on the low concentration sample using the reduced hyper-volume of parameters obtained from the full optimization sequence 1) on the same sample or 2) on the high concentration sample. When using the reduced hyper-volume from the same sample, the axonal periodical actin cytoskeleton was clearly visible on 48 out of 50 images (96 %). However, when using the reduced hyper-volume from the high concentration sample, imaging of the low concentration sample resulted in only 22 out of 50 (44 %) images on which the actin rings were detected. With this experiment we now show that using inadequate pre-settings can lead to a wrong evaluation on the occurrence of a given biological structure (Supplementary Fig. 32 and Results section 2, p. 7-8).

Unfortunately for the authors the most significant aspect of a STED microscope which varies over-time is the alignment of the optics which generate the excitation psf and super-impose the depletion donut over the emitted fluorescence. Clearly their system at present does not allow automated control of the hardware elements which can affect this alignment. It would be fully advantageous to have a system which can optimise its alignment, whereas changing the laser power is quite subjective in terms of the experimental outcome. I think this point reduces the impact of the paper.

The alignment of optics with automated routine is an already well established approach and already implemented in some commercial STED microscope (e.g. Leica TCS SP8 STED). The main cause of beam misalignment is generally due to unstable environmental conditions (heat, humidity, ventilation, vibrations). Nowadays, STED microscopes are much more stable (as long as they are kept in a stable environment). In our case, the STED microscope (Abberior STED expert line) used is fully stable without the need for realignment over many days. As cited in the introduction, offline optimization tools are available in the context of adaptive optics to correct for sample related aberrations. Providing solutions to stabilize the optics of a STED microscope is outside of the scope of our study and would not add an innovative contribution since this is already integrated in commercial microscopes.

The authors feel differently however: “Considering the high degree of variability in the expression level of the transfected protein, it is an important advantage to conduct the optimisation simultaneously to the imaging routine to ensure comparable results across experiments.”. This is a contentious topic however and needs consideration on a case-by-case basis. In the case of their GCAMP response I think they are justified, but such an approach should be used with extreme caution.

As described above, we performed an additional extensive series of experiments to demonstrate that online optimization is beneficial, especially in the context of live cell imaging with fluorescent proteins. To this extent, we transfected various proteins (PSD95-FingR, CaMKII, LifeAct, plasma membrane targeting sequence) all tagged with the broadly used GFP. We compared the optimization results of those proteins in 3 different cell types (hippocampal neurons, HEK293 and PC12 cells) and between different days and cell culture preparations. We show (Fig. 3 and Supplementary Fig. 18, 19, 21, 22, 24, 25, 26, 27, 29, 30) that the optimal imaging parameters vary greatly with the protein of interest, the cell type and the chosen cell. Prior optimization with GATTA-Beads as a standard or a given structure (here PSD95-FingR-GFP) led to very variable and unpredictable results in term of the targeted objectives (Supplementary Fig. 18, 19, 21, 22, 24, 25, 26, 27, 29, 30). These results support our conclusion that our proposed method is a valuable tool in the context of super-resolution imaging of living cells to adapt to biological variability and therefore obtain higher quality images.

Minor points

The paper is well written, but some of the sentences are little confusing due to repetition of nouns and lack of appropriate punctuation.

“This resulted in a higher sampling rate for the excitation power values (4.0 to 9.7 μ W) that are susceptible of maximizing the AA (Fig. 1d and Supplementary Fig. 4).”

“A replay experiment consists in simulating an experiment using previously acquired data in order to compare different approaches on the same data.”

“This simulates the imaging process by giving access to similar images to all compared approaches given similar behaviors.”

“For all fluorophores, the optimization process allowed to reach an image quality above 70%.”

We corrected these sentences and revised the manuscript for grammar and punctuation.

The paper is mostly well-presented. I found the figures with circles on quite confusing however e.g. Fig. 2b. Not sure if their maybe a better representation.

We considered alternative representations (heat maps, 3D graph) than the circles, but came back to it, as it helps bringing several features in the same plot (more than one optimization sequence per graph, number of sampled combinations). To help the reader interpret these plots, we added information in the figure and caption.

REVIEWERS' COMMENTS:

Reviewer #1 (Remarks to the Author):

I acknowledge the authors' efforts to address my comments regarding the figures and various clarifications in the manuscript.

However, my main concerns were about the "how to". While I could access the web page of the project (<http://www.optim-nanoscopy.net>), which contains almost no relevant information, all three Github links gave me 404 error, hence I was not able to confirm that the authors complied in providing open-source code and proper documentation for their project.

UPDATE 2018-09-11:

I finally accessed the Github pages and confirm the code is out there and properly documented, so I give my blessings to the authors for this article.

Reviewer #2 (Remarks to the Author):

First of all I would thank the authors of the manuscript to take into account the comments of the review so carefully. I really think that most of the issues raised from the previous version of the manuscript have been solved. This version, in my opinion, is much more clear and provides a better vision of the problem and solution presented there. As I mentioned in my previous review, I really think that the problem presented here is actually a limitation in the usage of SR techniques (as well as multimodal imaging) and several non-experts have limited motivation to explore the possibilities provided by the SR.

I have, however, two minor comments.

Firstly I think that it should be included in the manuscript a comment or mention of the limitation of using this approach in 'closed' microscopes. In this manuscript the microscope used is an Abberior STED system that includes the powerful option of scripting, which is not easily accessible in other systems. Thus, the application of such methodology, in very practical terms, is quite restricted to the systems that have that implemented or are custom made. I think that a comment in this direction could illustrate the need of 'opening' the access to scripting on the very closed microscopes.

The second comment is about the 'prior knowledge' comparison. I agree that reducing the hyper-volume of solutions could lead to 'skip' possible better solutions. However I think that the comparison is a bit biased. If the search of solutions in the low intensity case is made in the reduced space of solution found in the case including both high and low labeling, indeed I think that this space would be too reduced because the variability on this space of low labeling is larger. In the other hand, probably if this search would be performed in the other scenario (high labeling) the space of parameters would be smaller. Because of that I speculate that the results would be more robust. This does not invalidate the issue to find the right size of hyper-volume and the risk of arbitrarily reduce it for searching. However, at least, I think that the authors should add a comment on this.

Apart from these two minor issues, I think that this manuscript rises an important issue on the SR parameters settings search affecting dramatically the usage of such techniques for producing very relevant information and, then, important research. The elegant solution presented to address the problem can be a first step to open the deep learning techniques to solve daily problems in research improving the research and saving efforts, so increasing the efficiency.

Due to all of that I would recommend the manuscript for publication.

Reviewer #3 (Remarks to the Author):

The authors have worked hard to address many of the concerns of the reviewers. I am not completely convinced this is a practical or very straight-forward solution to the problem, but it is certainly a step in the right direction. Furthermore, I believe that in general auto-optimisation based on appearance of a biological specimen is a dangerous avenue to proceed along. The authors rebuttal to this sentiment: "One could reverse the argument by considering that a low quality (or low intensity) staining combined with non-suitable acquisition parameters could lead to a wrong interpretation of the biological information. ". In my opinion, too often we proceed with experimentation even if it pushes the boundaries of whats possible with the methodology. Your statement implies that your method is the lesser of two wrongs, but really we should only be proceeding with experiments if we can be sure what we are doing is right. I think the real place for auto-optimisation is to act has a validation of human activity, to caution them on a particular experiment, rather than to take over the whole procedure.

Additionally the authors have missed the point with my statement:

"the most significant aspect of a STED microscope which varies over-time is the alignment of the optics which generate the excitation psf and super-impose the depletion donut over the emitted fluorescence."

by saying: "The alignment of optics with automated routine is an already well established approach and already implemented in some commercial STED microscope (e.g. Leica TCS SP8 STED). ". Yes, one calibrates their system at the start of the day (or less frequently with a v.stable system), but this alignment procedure does not take care of sample specific aberrations and other artefacts concerned with exploring a 3-D sample with depth and inconsistencies in its refractive index. This is an area of continual research. I think the use, and optimization, of adaptive optics for this approach would be a clear benefit.

In summary, I think the authors have done enough to justify their paper in Nature Communications. I still have some reservations, but I believe this paper is a valuable contribution toward this contentious field and am happy to recommend it for publication.

Second review process

Reviewer #1 (Remarks to the Author):

I acknowledge the authors' efforts to address my comments regarding the figures and various clarifications in the manuscript.

However, my main concerns were about the "how to". While I could access the web page of the project (<http://www.optim-nanoscopy.net>), which contains almost no relevant information, all three Github links gave me 404 error, hence I was not able to confirm that the authors complied in providing open-source code and proper documentation for their project.

UPDATE 2018-09-11:

I finally accessed the Github pages and confirm the code is out there and properly documented, so I give my blessings to the authors for this article.

We thank the reviewer for his constructive comments. The error message obtained when trying to access the repositories may arrive when the users are not already connected to github. To prevent this from happening, users should connect to github with the appropriate credentials before clicking on the hyperlinks for private repositories. The repositories related to the paper will be public upon publications. We will also update the web page with more relevant information upon publication.

Reviewer #2 (Remarks to the Author):

First of all I would thank the authors of the manuscript to take into account the comments of the review so carefully. I really think that most of the issues raised from the previous version of the manuscript have been solved. This version, in my opinion, is much more clear and provides a better vision of the problem and solution presented there. As I mentioned in my previous review, I really think that the problem presented here is actually a limitation in the usage of SR techniques (as well as multimodal imaging) and several non-experts have limited motivation to explore the possibilities provided by the SR.

I have, however, two minor comments.

Firstly I think that it should be included in the manuscript a comment or mention of the limitation of using this approach in 'closed' microscopes. In this manuscript the microscope used is an Abberior STED system that includes the powerful option of scripting, which is not easily accessible in other systems. Thus, the application of such methodology, in very practical terms, is quite restricted to the systems that

have that implemented or are custom made. I think that a comment in this direction could illustrate the need of 'opening' the access to scripting on the very closed microscopes.

The second comment is about the 'prior knowledge' comparison. I agree that reducing the hyper-volume of solutions could lead to 'skip' possible better solutions. However I think that the comparison is a bit biased. If the search of solutions in the low intensity case is made in the reduced space of solution found in the case including both high and low labeling, indeed I think that this space would be too reduced because the variability on this space of low labeling is larger. In the other hand, probably if this search would be performed in the other scenario (high labeling) the space of parameters would be smaller. Because of that I speculate that the results would be more robust. This does not invalidate the issue to find the right size of hyper-volume and the risk of arbitrarily reduce it for searching. However, at least, I think that the authors should add a comment on this.

Apart from these two minor issues, I think that this manuscript rises an important issue on the SR parameters settings search affecting dramatically the usage of such techniques for producing very relevant information and, then, important research. The elegant solution presented to address the problem can be a first step to open the deep learning techniques to solve daily problems in research improving the research and saving efforts, so increasing the efficiency. Due to all of that I would recommend the manuscript for publication.

We thank the reviewer for his constructive comments. We included one sentence mentioning the scripting option of the Abberior microscopes in the Methods section. We further modified the end of the paragraph about the choice of hyper-volume of solutions (p. 7) to better address the issue raised by the reviewer.

Reviewer #3 (Remarks to the Author):

The authors have worked hard to address many of the concerns of the reviewers. I am not completely convinced this is a practical or very straight-forward solution to the problem, but it is certainly a step in the right direction. Furthermore, I believe that in general auto-optimisation based on appearance of a biological specimen is a dangerous avenue to proceed along. The authors rebuttal to this sentiment: "One could reverse the argument by considering that a low quality (or low intensity) staining combined with non-suitable acquisition parameters could lead to a wrong interpretation of the biological information. ". In my opinion, too often we proceed with experimentation even if it pushes the boundaries of whats possible with the methodology. Your statement implies that your method is the lesser of two wrongs, but really we should only be proceeding with experiments if we can be sure what we are doing is right. I think the real place for auto-optimisation is to act has a validation of human activity, to caution them on a particular experiment, rather than to take over the whole procedure.

We agree with the reviewer. As described in the paper, depending on the experiment, optimization with an user in the loop can be chosen rather than the fully-automated platform. It is also possible to include other types of evaluation metrics depending on the biological question to replace the quality assessment if necessary.

Additionally the authors have missed the point with my statement:

"the most significant aspect of a STED microscope which varies over-time is the alignment of the optics which generate the excitation psf and super-impose the depletion donut over the emitted fluorescence."

by saying: "The alignment of optics with automated routine is an already well established approach and already implemented in some commercial STED microscope (e.g. Leica TCS SP8 STED). ". Yes, one calibrates their system at the start of the day (or less frequently with a v.stable system), but this alignment procedure does not take care of sample specific aberrations and other artefacts concerned with exploring a 3-D sample with depth and inconsistencies in its refractive index. This is an area of continual research. I think the use, and optimization, of adaptive optics for this approach would be a clear benefit.

We thank the reviewer for the clarification about the alignment of optics in a STED microscope. We added a sentence in the discussion to refer to the use of adaptive optics for offline optimization of confocal microscopy.

In summary, I think the authors have done enough to justify their paper in Nature Communications. I still have some reservations, but I believe this paper is a valuable contribution toward this contentious field and am happy to recommend it for publication.